# ASK YOUR DISTRIBUTION SHIFT IF PRE-TRAINING IS RIGHT FOR YOU

## ABSTRACT

Pre-training is a widely used approach to develop models that are robust to distribution shifts. However, in practice, its effectiveness varies: fine-tuning a pre-trained model improves robustness significantly in some cases but *not at all* in others (compared to training from scratch). In this work, we seek to characterize the failure modes that pre-training *can* and *cannot* address. In particular, we focus on two possible failure modes of models under distribution shift: poor extrapolation (e.g., they cannot generalize to a different domain) and biases in the training data (e.g., they rely on spurious features). Our study suggests that, as a rule of thumb, pre-training can help mitigate poor extrapolation but not dataset biases. After providing theoretical motivation and empirical evidence for this finding, we explore two of its implications for developing robust models: (1) pre-training and interventions designed to prevent exploiting biases have complementary robustness benefits, and (2) fine-tuning on a (very) small, non-diverse but *de-biased* dataset can result in significantly more robust models than fine-tuning on a large and diverse but biased dataset.

## 1 INTRODUCTION

A common paradigm for developing machine learning models is pre-training them on a large, diverse dataset (e.g., ImageNet (Deng et al., 2009), JFT-300M (Sun et al., 2017), LAION-5B (Schuhmann et al., 2022)) and then fine-tuning them on task-specific data. Indeed, compared to training from scratch, fine-tuning a pre-trained model often significantly improves performance and reduces computational costs (Sharif Razavian et al., 2014; Sun et al., 2017; Kornblith et al., 2019).

Yet another benefit that pre-training may offer is *distribution shift robustness*. Specifically, machine learning models tend to suffer from distribution shifts, i.e., changes between the *reference distribution* used to develop the model and the *shifted distribution* that the model actually encounters when deployed. For example, a tumor identification model trained on tissue slide images from one hospital might perform poorly when deployed at another hospital (Bandi et al., 2018; Koh et al., 2020). Notably, different models (with different architectures, hyperparameters, etc.) tend to be similarly sensitive to a given distribution shift. However, models pre-trained on auxiliary data and then fine-tuned on the reference distribution can break this trend, exhibiting substantially higher performance on the shifted distribution than models trained from scratch with the same performance on the reference distribution (Taori et al., 2020; Miller et al., 2020; 2021).

These robustness benefits of pre-training are promising, but they are *not* universal. In particular, fine-tuning the same pre-trained model can yield significant robustness gains on some distribution shifts but not on others (Radford et al., 2021; Wortsman et al., 2021). To attain robustness to the latter shifts, would fine-tuning a larger model pre-trained on more data suffice? Or are there fundamental limitations to the robustness that pre-training can provide? To answer these questions, we develop a more fine-grained understanding of how pre-training improves robustness. Specifically, we ask:

*Can we identify and characterize the failure modes that pre-training can and cannot address?*

Recall that under distribution shift, models can fail in a number of ways. One of them is their inability to *extrapolate* effectively outside of the reference distribution (Gulrajani & Lopez-Paz, 2020; Koh et al., 2020). If, for instance, a model is trained only on photos taken during the day, then this model might fail when deployed on photos taken at night.

Models can also underperform even when the shifted distribution does not contain anything "new." In particular, they can fail due to *biases* in the reference distribution. For example, if a certain feature is spuriously correlated with the label in the reference distribution, a model might learn to exploit this relationship and fail on examples encountered during deployment where it does not hold (Arjovsky et al., 2019; Geirhos et al., 2020).

## 1.1 OUR CONTRIBUTIONS

To identify the failure modes that pre-training can address, we study the robustness benefits of pre-training under two types of distribution shifts: (1) shifts where extrapolation is necessary and (2) shifts where extrapolation is not needed. We start by analyzing a simple logistic regression setting and illustrate why pre-training might improve robustness to the former type of shift, but not the latter (Section 3). We subsequently build on this intuition by measuring the robustness benefits of pre-training on synthetic and natural distribution shifts of each type (Section 4). Our results suggest the following rule of thumb: pre-training helps specifically with extrapolation, but does not address other failures, for example, those stemming from dataset biases.

**Implications for developing robust models.** Guided by this rule of thumb, we explore two related avenues for harnessing pre-training to develop robust models.

1. *Combining pre-training with interventions designed to handle bias* (Section 5): There are a number of robustness interventions specifically designed to mitigate biases present in a training dataset (Byrd & Lipton, 2019; Sagawa et al., 2020a; Liu et al., 2021; Kirichenko et al., 2022; Idrissi et al., 2022). Our findings suggest that pre-training and this kind of intervention address two different sources of failures (the former helping with extrapolating and the latter with avoiding dataset biases) and thus may be viewed as complementary. We indeed find that combining them can yield models with both sets of benefits.

2. *Curating datasets for fine-tuning* (Section 6): One possible intervention that aims to address dataset biases is curating a de-biased dataset. In general, however, the de-biasing process might be prohibitively expensive. That said, we find that if we leverage pre-training to help with extrapolation, we might only need a small, non-diverse fine-tuning dataset; such a dataset might actually be feasible to de-bias. Specifically, we demonstrate that fine-tuning on a hair color classification dataset with only 64 examples that was carefully de-biased yields greater robustness than fine-tuning on the entire CelebA dataset (Liu et al., 2015).

## 2 BACKGROUND

**Fine-tuning a pre-trained model.** Methods for fine-tuning a pre-trained model vary: two common strategies are *full fine-tuning*, in which one continues training the entire model, and *linear probing*, in which one only fine-tunes the final layer. Some recent pre-trained models with natural language supervision (e.g., CLIP (Radford et al., 2021), ALIGN (Jia et al., 2021)) can also be applied to a downstream task in a *zero-shot* context (i.e., without fine-tuning) by specifying the task through a text description. In this work, we focus on the full fine-tuning strategy, which typically outperforms linear probing and zero-shot models on the reference distribution. We also will sometimes consider linear probing followed by full fine-tuning (LP-FT), which can in some cases improve over full fine-tuning alone in terms of performance and robustness (Kumar et al., 2022). We discuss other fine-tuning strategies in Appendix D.1.

**Measuring robustness.** For many distribution shifts, different models trained from scratch on the reference distribution exhibit similar degrees of robustness to the shift. Specifically, when varying architectures, hyperparameters and training methods there is often a strong *linear* relationship between the *reference accuracy* and *shifted accuracy*[1] (i.e., the accuracies on the reference and shifted distributions, respectively) (Taori et al., 2020; Miller et al., 2020; 2021). This relationship, dubbed *accuracy on the line*, can be visualized by plotting shifted accuracies against reference accuracies and finding a linear fit. When this linear trend is strong (i.e., shifted accuracies are highly correlated with reference accuracies), one can predict the shifted accuracy of models trained from scratch from their reference accuracy. Furthermore, to quantify the robustness of a model trained with a

---

[1]For a linear relationship, accuracies are *probit-scaled* (transformed by the inverse of the Gaussian CDF).

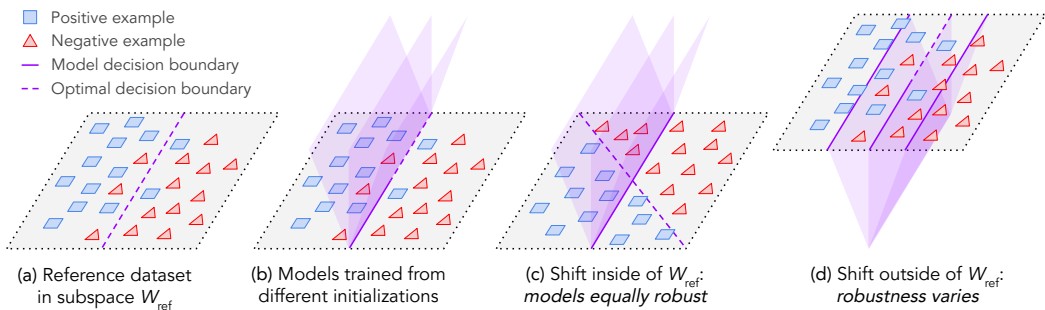

Figure 1: **Illustration of logistic regression setting.** (a) Consider a reference dataset that lies within a subspace $W_{\text{ref}}$ of $\mathbb{R}^d$. (b) Models trained from different initializations all learn the same (optimal) decision boundary in $W_{\text{ref}}$, but may behave differently outside of $W_{\text{ref}}$. (c) Under shifts within $W_{\text{ref}}$, models with different initializations are equally robust. (d) Under shifts outside of $W_{\text{ref}}$, initialization can affect robustness.

robustness intervention beyond the "baseline" of models trained from scratch, one can measure the amount by which its shifted accuracy exceeds the linear fit's prediction, a metric known as *effective robustness* (ER) (Taori et al., 2020). In this work, we choose to study distribution shifts for which accuracy on the line holds (i.e., the linear fit is strong) and quantify robustness by computing effective robustness (see, e.g., Figure 3). See Appendix B.1.2 for additional details.

## 3 STUDYING PRE-TRAINING IN A LOGISTIC REGRESSION SETTING

Our central goal is to understand the failure modes that pre-training *can* and *cannot* address. To this end, we first study the robustness benefits of pre-training in a simple logistic regression setting (illustrated in Figure 1).

**Setup.** Suppose that we are given access to a reference dataset $S_{\text{ref}}$ of input-label pairs, each consisting of a $d$-dimensional input $x \in \mathbb{R}^d$ and a binary label $y \in \{-1, 1\}$. We learn a linear model for this task by finding weights $w \in \mathbb{R}^d$ that minimize the (standard) logistic loss on $S_{\text{ref}}$:

$$L_{\text{ref}}(w) = \sum_{(x,y) \in S_{\text{ref}}} \log(1 + e^{-w^\top x \cdot y}). \tag{1}$$

We assume that the reference dataset $S_{\text{ref}}$ satisfies the following conditions:

1. **Inputs in $S_{\text{ref}}$ lie within a $k$-dimensional (with $k < d$) subspace $W_{\text{ref}}$ of $\mathbb{R}^d$.** Intuitively, this condition corresponds to features lacking certain variation in the reference dataset.

2. **The logistic loss $L_{\text{ref}}$ has a minimum value.** This condition ensures that minimizing $L_{\text{ref}}$ is well-defined. Note that there may be multiple weights that attain this minimum value.

Starting with initial weights (which, in our case, are either random or the result of pre-training), suppose that we use gradient descent to minimize $L_{\text{ref}}(w)$. We would like to understand how well the resulting model performs under distribution shift. To do so, we establish the following theorem (proof in Appendix A):

**Theorem 3.1.** *Suppose that we start with initial weights $w_{init} \in \mathbb{R}^d$ and run gradient descent to minimize $L_{ref}(w)$. With an appropriately chosen learning rate, gradient descent converges to weights $\hat{w}$ that minimize $L_{ref}$. Letting $proj_{W_{ref}} w_{init}$ be the projection of $w_{init}$ onto $W_{ref}$, $\hat{w}$ can be written as*

$$\hat{w} = w_{ref}^* + (w_{init} - proj_{W_{ref}} w_{init}) \tag{2}$$

*where $w_{ref}^*$ is a property of the reference dataset $S_{ref}$ and lies within the reference subspace $W_{ref}$.*

Theorem 3.1 enables us to decompose the learned model's weights $\hat{w}$ into two terms: $w_{\text{ref}}^*$ and $(w_{\text{init}} - \text{proj}_{W_{\text{ref}}} w_{\text{init}})$. Notice that the first term is just a property of the reference dataset and is in the reference subspace $W_{\text{ref}}$, while the second term depends on $w_{\text{init}}$ and is *orthogonal* to $W_{\text{ref}}$. As a result, the reference dataset itself fully specifies the model's behavior on inputs in $W_{\text{ref}}$, while the

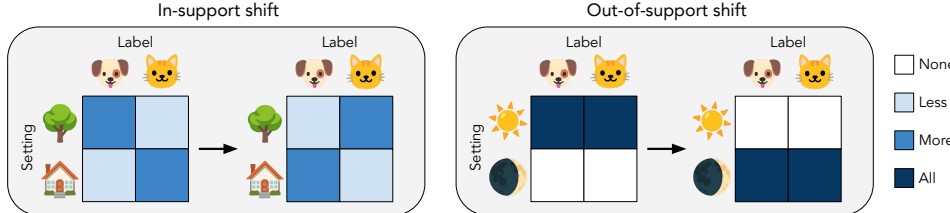

Figure 2: **Examples of in-support and out-of-support shifts.** One example of an *in-support shift* (left) is a shift in which the indoor/outdoor frequencies of animal appearances change, but the possible combinations of animal and setting remain the same. An example of an *out-of-support shift* (right) is a shift from day to night: the nighttime setting is entirely novel.

initialization determines how the model extends outside of $W_{\text{ref}}$. Consequently, changing a model's initialization (e.g., with pre-training) can affect performance outside of $W_{\text{ref}}$, but not within $W_{\text{ref}}$.

This observation suggests the following key intuition: pre-training can improve robustness to a distribution shift *only* when the shifted distribution contains "out-of-support" inputs, that is, inputs that could not be reasonably sampled from the reference distribution. In other words, pre-training helps specifically with extrapolation outside of the reference distribution.

## 4 Exploring the Empirical Robustness Benefits of Pre-Training

In Section 3, we found that in a simple logistic regression setting, pre-training helps *specifically* with extrapolation. We now want to assess whether this principle holds more broadly. To do so, we measure the robustness benefits of pre-training under two types of shifts: *in-support shifts*, where models *cannot* fail due to poor extrapolation (but might fail for other reasons, e.g., dataset biases), and *out-of-support shifts*, where models *can* fail due to poor extrapolation (see Figure 2). We begin by describing these two types of shifts in more detail and providing intuitions for why pre-training might improve robustness to out-of-support shifts, but not in-support shifts.

**In-support shift.** A distribution shift is *in-support* if any input that could be sampled from the shifted distribution could also be reasonably sampled from the reference distribution. In other words, the shifted distribution does not contain anything "new"; however, an in-support shift can still cause failures if, for example, the reference distribution is *biased*. To illustrate this failure mode, consider a cat vs. dog image classification task in which photos are either taken indoors or outdoors. Suppose that in the reference distribution 90% of cats appear indoors and 90% of dogs appear outdoors (i.e., the setting is spuriously correlated with the animal). A model trained on this distribution would likely rely (at least in part) on indoor vs. outdoor features (Xiao et al., 2020; Geirhos et al., 2020). Thus, under a shift in which the setting/animal correlation is reversed (which would be in-support but out-of-distribution), the model would likely underperform. If pre-training helps specifically with extrapolation, then it would not address this failure mode and, more generally, could not improve robustness to in-support shifts.

**Out-of-support shift.** A distribution shift is *out-of-support* if there exists an input that could be sampled from the shifted distribution but could not be reasonably sampled from the reference distribution. For example, consider a cat vs. dog image classification task in which photos from the reference distribution are taken during the day and photos from the shifted distribution are taken at night. In this case, the shifted distribution contains images with previously unseen lighting conditions. Here, a model trained from scratch might learn features that are sensitive to lighting and thus fail under the shift. Meanwhile, pre-training could provide priors for extrapolating, e.g., by producing features that are agnostic to lighting conditions as a starting point, leading to greater robustness.

### 4.1 Constructing synthetic in-support and out-of-support shifts

We now want to measure the robustness gains that pre-training provides on in-support and out-of-support shifts. To this end, we explicity construct two shifts of each type by modifying CIFAR-10 (Krizhevsky, 2009): (1) a "tint shift" in which we add a tint that is spuriously correlated with the label in the reference dataset, but not in the shifted dataset, (2) a "label shift" in which the relative

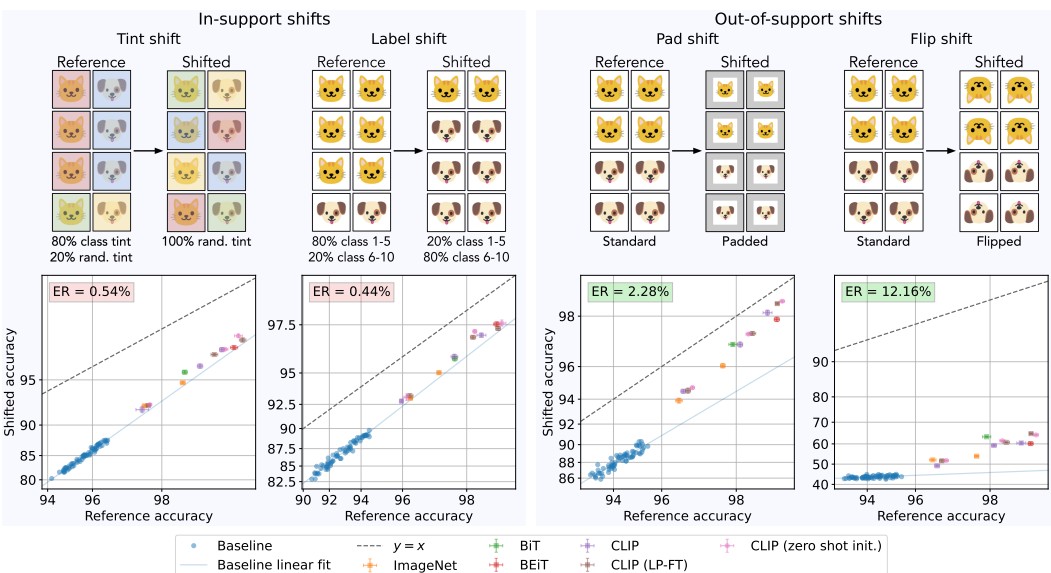

Figure 3: **Robustness of pre-trained models to synthetic in-support and out-of-support shifts.** For each of two in-support shifts (left) and two out-of-support shifts (right) constructed by modifying CIFAR-10, the reference and shifted accuracies of models trained from scratch (in blue) are linearly correlated. Pre-trained models exhibit little effective robustness (i.e., little improvement over the linear trend, see Section 2) on the in-support shifts, but have significant effective robustness on the out-of-support shifts (averages in the top left of each plot). Error bars denote 95% confidence intervals over 8 random trials. See Appendix B.2 for descriptions of the individual models.

frequencies of classes change between the reference and shifted datasets, (3) a "pad shift" in which we pad images with a black border in the shifted dataset, and (4) a "flip shift" in which we vertically flip images in the shifted dataset (see the top of Figure 3 for visualizations).

For each shift, as a baseline, we train ResNet-18 (He et al., 2015) models from scratch on differently sized subsets of the reference dataset. Next, we fine-tune various pre-trained models and measure their effective robustness above this baseline. We select fine-tuning hyperparameters that maximize accuracy on the reference distribution (in Appendix C.1.1, we find that other reasonable hyperparameter choices yield similar robustness). See Appendix B.2 for a description of the exact setup.

We observe that pre-trained models exhibit substantial effective robustness on out-of-support shifts, but have very limited (yet non-zero) effective robustness on in-support shifts (see Figure 3). This broadly supports our hypothesis. However, note that if pre-training indeed only helped with extrapolation, then we might initially expect that it would yield *no* effective robustness to in-support shifts at all. We investigate this limited effective robustness of pre-trained models on in-support shifts in Appendix C.1.2 and conclude that it likely also derives from better extrapolation.

## 4.2 DIVIDING NATURAL SHIFTS INTO IN-SUPPORT AND OUT-OF-SUPPORT SPLITS

So far, we have constructed synthetic in-support and out-of-support shifts and observed that pre-training can significantly improve robustness to the latter but not the former. Now, we demonstrate that this principle seems to extend to natural shifts as well. Note that it is hard to find natural shifts that are "purely" in-support. After all, under natural shifts the shifted dataset may contain some inputs that are similar to those in the reference dataset and some that are not. For example, in a shift from photos to sketches, some sketches may look more photorealistic but most would probably be clearly distinguishable from photos. To measure robustness to each type of shift, we thus *divide* several natural shifted datasets each into an "in-support split" containing inputs that look like they could have come from the reference dataset and an "out-of-support split" containing the remaining inputs. We do so by training a classifier to distinguish between the reference and shifted datasets and using this classifier to approximate the probability of sampling a given shifted example from the reference distribution (see Appendix B.3.1 for the details of the splitting method).

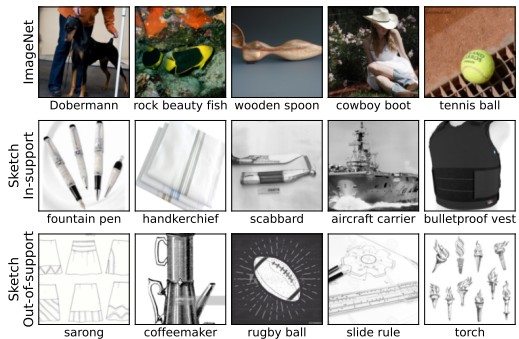

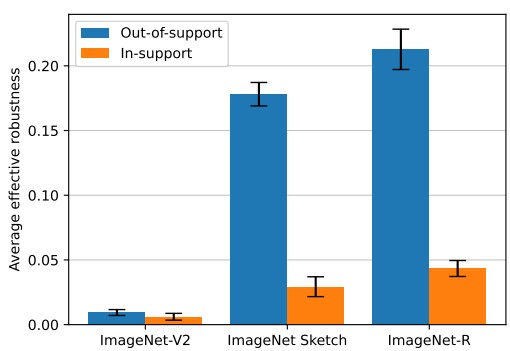

(a) Random samples from ImageNet (top) as well as the in-support split (middle) and out-of-support split (bottom) of ImageNet Sketch.

(b) Average effective robustness of 24 pre-trained models on each split of each of the three shifts. Error bars denote 95% confidence intervals.

Figure 4: We divide each of the ImageNet-V2, ImageNet Sketch and ImageNet-R datasets into an in-support split containing examples that look like ImageNet examples and an out-of-support split containing examples that look unlike ImageNet examples. We display samples from each split of ImageNet Sketch in Figure 4a and report the average effective robustnesses of pre-trained models in Figure 4b. See Appendix C.2.3 for scatterplots of reference vs. shifted accuracy.

Specifically, in our study, we consider three natural shifts of the ImageNet dataset: ImageNet-V2 (Recht et al., 2019), which closely resembles ImageNet, ImageNet Sketch (Wang et al., 2019), which consists of sketches of ImageNet classes, and ImageNet-R (Hendrycks et al., 2020a), which consists of "renditions" (e.g, paintings, sculptures, cartoons) of a subset of ImageNet classes. We choose these shifted datasets because they include many inputs that look like they could have come from ImageNet and many that do not (according to our splitting method)[2]. In Figure 4a, we visualize examples from the in-support and out-of-support splits of ImageNet Sketch.

Consistently with our hypothesis that pre-training helps specifically with extrapolation, on the out-of-support splits of ImageNet Sketch and ImageNet-R pre-trained models have substantially higher effective robustness than on the respective in-support splits (see Figure 4b). On both ImageNet-V2 splits, however, pre-trained models have very little effective robustness. This may be because ImageNet-V2 is visually similar to ImageNet, so poor extrapolation might not be a significant failure mode (instead, the performance drop may be due to an increased presence of "harder" examples, as Recht et al. (2019) suggest). Thus, if pre-training helps only with extrapolation, it would not be able to substantially improve robustness on the ImageNet-V2 out-of-support examples. See Appendix B.3.2 for a description of the exact setup.

## 5 COMBINING PRE-TRAINING WITH INTERVENTIONS FOR HANDLING BIAS

Our observations in Section 4 suggest that pre-training indeed helps prevent failures caused by poor extrapolation but not those stemming from biases in the reference dataset. How, then, can we develop models that avoid *both* failure modes? In this section, we explore one possible strategy: combining pre-training with interventions specifically designed to handle dataset biases.

In particular, we investigate the effectiveness of this strategy on WILDS-FMoW (Christie et al., 2018; Koh et al., 2020), a distribution shift benchmark for classifying satellite images (in Appendix C.3.1, we provide a similar analysis for a synthetic distribution shift). In WILDS-FMoW, the reference dataset consists of satellite images taken between 2002 and 2012, while the shifted dataset consists of satellite images taken between 2016 and 2017. Additionally, the images depict different regions and models typically underperform on underrepresented regions. Following Koh et al. (2020), we evaluate the *worst-group accuracy* (the minimum accuracy across groups—in our case, regions) on the shifted dataset. Hence, robustness to this shift requires being able to both extrapolate

---

[2]We also explored ObjectNet (Barbu et al., 2019) and ImageNet-Vid-Robust (Shankar et al., 2019) but our splitting method marks fewer than 50 examples from these shifted datasets as "in-support," and thus we cannot reliably measure in-support accuracy.

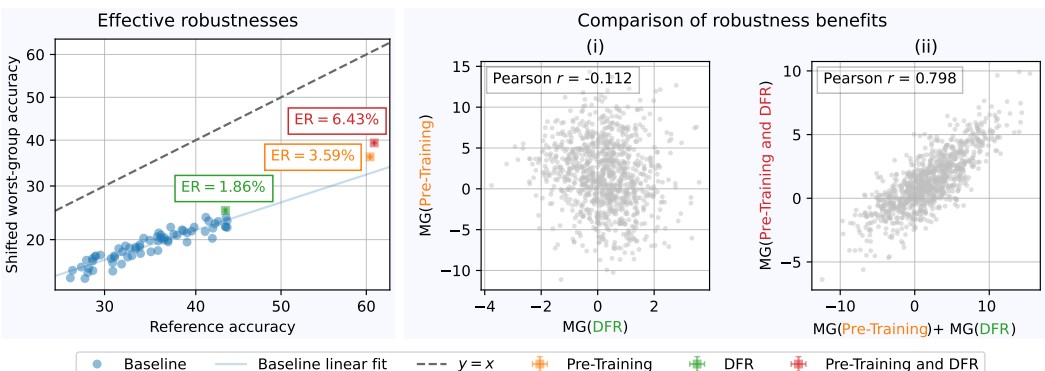

Figure 5: **Combining pre-training and *Deep Feature Reweighting* (DFR) on the WILDS-FMoW distribution shift.** Pre-training and DFR (an intervention designed to handle dataset biases (Kirichenko et al., 2022)) each yield some effective robustness (ER) and combining these two interventions yields the most effective robustness (left). The margin gains (MG) of pre-training and DFR on the shifted dataset are *negatively* correlated (right, i), indicating that they improve performance on *different* subpopulations. Meanwhile, the margin gains of using both interventions are highly correlated with the *summed* margin gains of using each individually (right, ii), suggesting that combining pre-training with DFR improves performance on *both* of these subpopulations. Error bars denote 95% confidence intervals over 64 random trials.

to later years *and* perform consistently across regions (e.g., by avoiding biases that are harmful to performance on some regions).

Aiming to overcome these two challenges, we leverage two types of interventions. To extrapolate better to later years, we initialize the model via pre-training; specifically, we obtain our model by fine-tuning a CLIP (Radford et al., 2021) ViT-B/32 (Dosovitskiy et al., 2021) model. To handle potential biases in the reference dataset, we employ *Deep Feature Reweighting* (DFR) (Kirichenko et al., 2022), an intervention intended to de-bias a model by re-training just the final layer on group-balanced data. We measure the effective robustness of each intervention over a baseline of ResNet-18 models trained from scratch (see Appendix B.4 for a description of the exact setup). We find that pre-training and DFR each yield some effective robustness and that combining the two yields greater effective robustness than applying either individually (see the left side of Figure 5).

**Understanding robustness benefits.** We observe that combining pre-training and DFR can be effective for developing robust models, but is this actually because they address different failure modes, as we suggest? To answer this question, we conduct a more fine-grained comparison of the robustness benefits of these two interventions.

If pre-training and DFR address different failure modes, then we might expect that they improve performance on different subpopulations. To quantify the extent to which a given intervention improves performance on a particular example, we measure the resulting (expected) increase in *correct-class margin*. The *correct-class margin* (a measure of confidence in the correct class) of a model $M$ on an example $(x, y)$ is defined as

$$f(M, (x, y)) = (\text{logit for correct class}) - (\text{highest incorrect logit}). \quad (3)$$

Here, we opt to measure the increase in correct-class margin instead of a more explicit performance metric (e.g., an indicator for correctness) in order to capture fine-grained differences in model behavior. Next, let $\mathcal{A}$ be a learning algorithm with the intervention of interest and $\mathcal{B}$ be a baseline learning algorithm (where $A \sim \mathcal{A}$ denotes that model $A$ is obtained from running $\mathcal{A}$). Then the increase in correct-class margin when applying the intervention, i.e., the *margin gain*, is

$$\text{MG}_{\mathcal{B}}(\mathcal{A}, (x, y)) = \mathbb{E}_{A \sim \mathcal{A}, B \sim \mathcal{B}}[f(A, x, y) - f(B, x, y)]. \quad (4)$$

Now, to determine whether pre-training and DFR improve performance on different subpopulations, we compute the *correlation* between their margin gains on the shifted dataset. We find that their margin gains are *negatively* correlated, suggesting that their benefits are indeed complementary. We

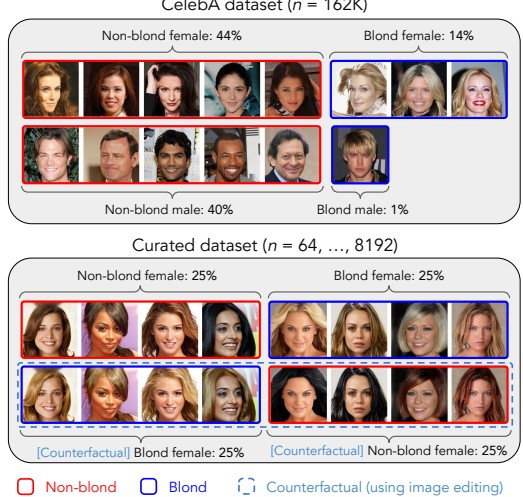
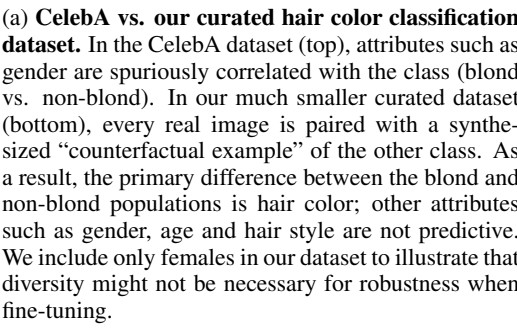

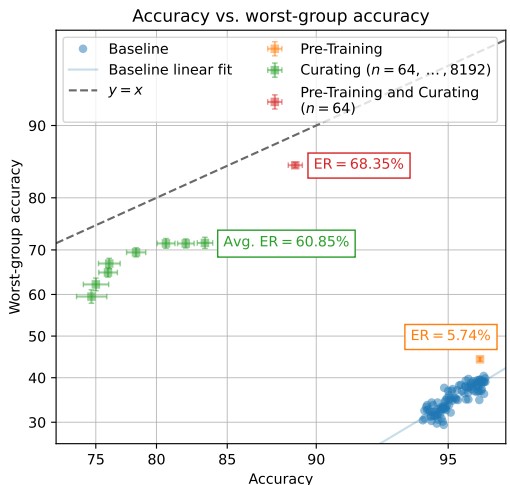

(a) **CelebA vs. our curated hair color classification dataset.** In the CelebA dataset (top), attributes such as gender are spuriously correlated with the class (blond vs. non-blond). In our much smaller curated dataset (bottom), every real image is paired with a synthesized "counterfactual example" of the other class. As a result, the primary difference between the blond and non-blond populations is hair color; other attributes such as gender, age and hair style are not predictive. We include only females in our dataset to illustrate that diversity might not be necessary for robustness when fine-tuning.

(b) **Fine-tuning on our curated dataset.** Fine-tuning a pre-trained model on the CelebA dataset (orange) yields little effective robustness over a baseline of models trained from scratch (blue). However, fine-tuning the same pre-trained model on just 64 examples from our curated dataset (red) yields a model with both high effective robustness and high accuracy. Training from scratch on our curated dataset (green) also yields high effective robustness, but results in substantially lower accuracy than pre-trained models, even with many more examples. Error bars denote 95% confidence intervals over 64 random trials.

Figure 6: Fine-tuning a pre-trained model on a small, non-diverse but de-biased dataset (see Figure 6a) yields a robust and performant model for hair color classification in CelebA (see Figure 6b).

also measure the correlation between the margin gains of combining pre-training with DFR and the *summed* margin gains of the individual interventions. We find that they are highly correlated, suggesting that combining pre-training and DFR not only yields high effective robustness but in fact leads to models with both sets of benefits (see the right side of Figure 5).

## 6 CURATING DATASETS FOR FINE-TUNING

In Section 5, we explored pairing pre-training with interventions specifically designed to address dataset biases. We observed that this strategy can be effective for developing models that both extrapolate effectively *and* avoid undesirable biases present in the reference distribution.

In this section, we highlight one such intervention: training on a carefully curated (and, in particular, de-biased) dataset *instead* of the original reference dataset. In general, de-biasing a large and diverse dataset may be prohibitively expensive. However, if we can rely on pre-training for extrapolation (as suggested in Section 4), we might only need a small, non-diverse fine-tuning dataset, which would be more feasible to de-bias. Thus, curating such a dataset and then fine-tuning a large pre-trained model on it might be a relatively inexpensive method for developing robust and performant models.

As a case study, we consider the task of predicting hair color (blond vs. non-blond) in the CelebA dataset (Liu et al., 2015). In this dataset, hair color is spuriously correlated with other attributes (especially gender). For example, 24% of females are blond, while only 2% of males are blond. Following works studying *group robustness* (Sagawa et al., 2020a; Liu et al., 2021; Kirichenko et al., 2022), we measure worst-group accuracy to assess robustness rather than measuring accuracy on an explicit shifted dataset. In this case, the four groups are blond females, non-blond females, blond males and non-blond males. A model exploiting the spurious correlation between gender and hair color would likely perform poorly on the underrepresented group of blond males.

**Curating a de-biased dataset.** To curate an de-biased dataset for hair color classification with $n$ examples, we construct a "counterfactual example" for each of $n/2$ CelebA examples by changing the person's hair to a color corresponding to the opposite class (i.e., blond to non-blond and vice versa). We ensure that attributes besides hair color remain unchanged and include both the original and edited images in our dataset. Hence, attributes that are spuriously correlated with hair color in the CelebA dataset (e.g., gender, age) are equally represented in the blond and non-blond populations of our curated dataset. To illustrate that this dataset does *not* need to be diverse to yield high robustness and performance when fine-tuning, we restrict the dataset to include *only* females. See Figure 6a for a visualization of the dataset and Appendix B.5 for the image editing process.

**Fine-tuning on a de-biased dataset.** As expected, models trained from scratch on the CelebA dataset exhibit high accuracy but very low worst-group accuracy, likely because they rely on gender to predict hair color (see Figure 6b). Furthermore, a pre-trained model fine-tuned on the CelebA dataset exhibits very little effective robustness above these models trained from scratch, consistently with our hypothesis that pre-training does not mitigate dataset biases. However, we observe that fine-tuning a pre-trained model on *just* 64 examples from our curated dataset yields a model with both high accuracy *and* effective robustness. Finally, we also train models from scratch on our curated dataset and find that they exhibit substantial effective robustness, but require many more examples to attain a comparable accuracy. This suggests that the extrapolation benefits of pre-training are key to make effective use of our small, non-diverse curated dataset. In particular, as we illustrate in Appendix C.4.1, pre-training improves extrapolation from the female-only curated dataset to males.

## 7 RELATED WORK

**Characterizing distribution shifts.** There exists a plethora of definitions for characterizing distribution shifts, many of which are aligned with the in-support and out-of-support chracterizations that we discuss in this work. For example, *domain generalization* involves shifts in which the reference and shifted distributions are from different domains (Koh et al., 2020; Gulrajani & Lopez-Paz, 2020). In a *subpopulation shift*, subpopulations appear with different frequencies in the reference and shifted distributions (Santurkar et al., 2021; Koh et al., 2020; Yang et al., 2023). In shifts with *spurious correlations*, certain features are predictive in the reference distribution but not in the shifted distribution (Arjovsky et al., 2019; Sagawa et al., 2020b). Two more formal characterizations are *covariate shift* (Shimodaira, 2000), under which $p(y|x)$ is fixed, and *label shift* (Lipton et al., 2018), under which the label distribution may change but $p(x|y)$ is fixed. We relate these definitons to in-support and out-of-support shifts in Appendix D.3.

**Robustness benefits of pre-training.** Several works have suggested that pre-training can be an effective strategy for improving robustness to distribution shifts (Hendrycks et al., 2019; 2020a;b; Tu et al., 2020; Wiles et al., 2021; Andreassen et al., 2021). In particular, Wiles et al. (2021) define different types of distribution shifts and find that pre-training frequently improves performance under these shifts, while most other interventions primarily help in specific settings. In the natural language processing setting, Tu et al. (2020) argue that when pre-training helps with spurious correlations, it is because pre-trained models can generalize better from the small number of counterexamples to these correlations; as we discuss in Appendix D.4, this is consistent with our intuition that pre-training helps specifically with extrapolation. Lastly, Bommasani et al. (2021) discuss failure modes that pre-training is unlikely to address including spurious correlations (both in pre-training and fine-tuning datasets) and extrapolation across time.

## 8 CONCLUSION

In this work, we study the failure modes that pre-training *can* and *cannot* address. Our findings suggest that pre-training can help mitigate failures caused by poor extrapolation (e.g., inability to generalize to a new domain) but might not address other failures, such as those stemming from dataset biases. In light of this observation, dataset biases present a fundamental limitation that cannot be overcome by additional pre-training data or larger models. We thus encourage practitioners not to treat pre-training as a panacea for robustness. Instead, they should consider the specific failures modes they might encounter (i.e., ask their distribution shift) to determine if pre-training can help.

## 9 REPRODUCIBILITY STATEMENT

Our work is a scientific investigation of the robustness benefits of pre-training and does not include novel methods. We provide relevant references to existing methods in the main paper and experiment details such as hyperparameters and preprocessing steps in the appendix. We intend to release source code for running the experiments in this work upon publication.

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

# A    THEORETICAL RESULTS

## A.1    PROOF OF THEOREM 3.1

**Setup.**    Suppose that we are given access to a reference dataset $S_{\text{ref}}$ of input-label pairs $(x, y)$, with $x \in \mathbb{R}^d$ and $y \in \{-1, 1\}$. We decide to learn a linear classifier for this task by finding a weight $w$ that minimizes the (standard) logistic loss on $S_{\text{ref}}$:

$$L_{\text{ref}}(w) = \sum_{(x,y) \in S_{\text{ref}}} \log(1 + e^{-w^\top x \cdot y}). \tag{1}$$

We assume that the reference dataset $S_{\text{ref}}$ satisfies the following conditions:

1. **Inputs in $S_{\text{ref}}$ lie within a $k$-dimensional (with $k < d$) subspace $W_{\text{ref}}$ of $\mathbb{R}^d$.** Intuitively, this condition represents a lack of variation in certain features in the reference dataset (e.g., if the reference dataset has fixed lighting conditions).

2. **The logistic loss $L_{\text{ref}}$ has a minimum value.** This condition ensures that minimizing $L_{\text{ref}}$ is well-defined, and means (roughly) that the two classes are not linearly separable. Note that there may be multiple weights that attain this minimum value.

**Theorem 3.1.** *Suppose that we start with initial weights $w_{init} \in \mathbb{R}^d$ and run gradient descent to minimize $L_{ref}(w)$. With an appropriately chosen learning rate, gradient descent converges to weights $\hat{w}$ that minimize $L_{ref}$. Letting $proj_{W_{ref}} w_{init}$ be the projection of $w_{init}$ onto $W_{ref}$, $\hat{w}$ can be written as*

$$\hat{w} = w_{ref}^* + (w_{init} - proj_{W_{ref}} w_{init}) \tag{2}$$

*where $w_{ref}^*$ is a property of the reference dataset $S_{ref}$ and lies within the reference subspace $W_{ref}$.*

To prove Theorem 3.1, we will first show that running gradient descent starting from an initialization within $W_{\text{ref}}$ always converges to the same weights $w_{\text{ref}}^*$. We will then show that running gradient descent starting from an arbitrary initialization has the same convergence behavior except for an "offset" term $(w_{\text{init}} - \text{proj}_{W_{\text{ref}}} w_{\text{init}})$ representing the component of the initialization that lies outside of $W_{\text{ref}}$.

### A.1.1    CONVEXITY AND SMOOTHNESS OF THE LOSS

We begin by providing the gradient and hessian of $L_{\text{ref}}$ and using these to establish convexity (Lemma A.1) and smoothness (Lemma A.2) properties of $L_{\text{ref}}$. The gradient of $L_{\text{ref}}$ is

$$\nabla L_{\text{ref}}(w) = \sum_{(x,y) \in S_{\text{ref}}} x \cdot y \cdot \frac{1}{1 + e^{w^\top x \cdot y}}. \tag{5}$$

The Hessian of $L_{\text{ref}}$ is

$$\nabla^2 L_{\text{ref}}(w) = \sum_{(x,y) \in S_{\text{ref}}} xx^\top \cdot \frac{1}{2 + e^{-w^\top x \cdot y} + e^{w^\top x \cdot y}} = X^\top D(w) X \tag{6}$$

where $X \in \mathbb{R}^{|S_{\text{ref}}| \times d}$ is the matrix of inputs in $S_{\text{ref}}$ and $D(w) \in \mathbb{R}^{|S_{\text{ref}}| \times |S_{\text{ref}}|}$ is the diagonal matrix with $D(w)_{ii} = \frac{1}{2 + e^{-w^\top x \cdot y} + e^{w^\top x \cdot y}}$. Note in particular that the non-zero elements of $D(w)$ are in $(0, 1/4)$.

**Lemma A.1.** *The loss $L_{ref}$ is (1) convex on $\mathbb{R}^d$, (2) strictly convex on $W_{ref}$, and (3) strongly convex on any closed convex subset of $W_{ref}$.*

*Proof.* According to Taylor's Theorem, for any $u, v \in \mathbb{R}^d$, there exists a $\alpha \in [0, 1]$ such that

$$L_{\text{ref}}(v) = L_{\text{ref}}(u) + \nabla L_{\text{ref}}(u)^\top (v - u) + \frac{1}{2} \cdot (v - u)^\top \nabla^2 L_{\text{ref}}(v + \alpha \cdot (v - u))(v - u). \tag{7}$$

1. **Convexity on $\mathbb{R}^d$.** To show that $L_{\text{ref}}$ is convex on $\mathbb{R}^d$, we need to show that

$$L_{\text{ref}}(v) \geq L_{\text{ref}}(u) + \nabla L_{\text{ref}}(u)^\top (v - u)$$

for any $u, v \in \mathbb{R}^d$. Using (7), it suffices to show that $a^\top \nabla^2 L_{\text{ref}}(w)a \geq 0$ for any $a \in \mathbb{R}^d$ and $w \in \mathbb{R}^d$. Recall from (6) that $\nabla^2 L_{\text{ref}}(w) = X^\top D(w)X$. Thus, we have

$$
\begin{aligned}
a^\top \nabla^2 L_{\text{ref}}(w)a &= a^\top X^\top D(w)Xa \\
&= \|D(w)^{1/2}Xa\|_2^2 \\
&\geq 0
\end{aligned}
$$

2. **Strict convexity on $W_{\text{ref}}$.** Next, to show that $L_{\text{ref}}$ is strictly convex on $W_{\text{ref}}$, we need to show that

$$
L_{\text{ref}}(v) > L_{\text{ref}}(u) + \nabla L_{\text{ref}}(u)^\top (v - u)
$$

for any $u, v \in W_{\text{ref}}$. Using (7), it suffices to show that $a^\top \nabla^2 L_{\text{ref}}(w)a > 0$ for any non-zero $a \in W_{\text{ref}}$ and $w \in W_{\text{ref}}$. We know that $a^\top \nabla^2 L_{\text{ref}}(w)a = \|D(w)^{1/2}Xa\|_2^2$. Since $D(w)$ is diagonal with positive entries along the diagonal, $\|D(w)^{1/2}Xa\|_2^2 > 0$ if and only if $Xa \neq 0$. Recall that $W_{\text{ref}}$ is the subspace spanning the rows of $X$. Hence, since $a$ is non-zero and is in $W_{\text{ref}}$, we know that $Xa \neq 0$.

3. **Strong convexity on any closed convex subset of $W_{\text{ref}}$.** Finally, to show that $L_{\text{ref}}$ is strongly convex on any closed convex subset $T$ of $W_{\text{ref}}$, we need to show that there exists an $m > 0$ such that

$$
L_{\text{ref}}(v) \geq L_{\text{ref}}(u) + \nabla L_{\text{ref}}(u)^\top (v - u) + \frac{m}{2}\|v - u\|_2^2
$$

for any $u, v \in T$. Using (7), it suffices to show that there exists an $m > 0$ such that $a^\top \nabla^2 L_{\text{ref}}(w)a > \frac{m}{2} \cdot \|a\|_2^2$ for any $a \in W_{\text{ref}}$ and $w \in T$. Making use of the fact that $T$ is closed, let $\lambda_{\min}$ be the minimum diagonal entry of $D(w)$ for $w \in T$, that is,

$$
\lambda_{\min} = \min_{w \in T} \min_{i \in \{1, \ldots, |S_{\text{ref}}|\}} D(w)_{ii}.
$$

Next, let $c_{\min}$ be the minimum value of $\|Xa\|_2^2$ over unit vectors $a$ in $W_{\text{ref}}$, that is,

$$
c_{\min} = \min_{a \in W_{\text{ref}}, \|a\|_2 = 1} \|Xa\|_2^2.
$$

We previously established that $Xa \neq 0$ for any non-zero $a \in W_{\text{ref}}$, which means that $c_{\min} > 0$. Finally, we conclude that for $m = 2 \cdot \lambda_{\min} \cdot c_{\min}$, $a^\top \nabla^2 L_{\text{ref}}(w)a = \|D(w)^{1/2}Xa\|_2^2 \geq \lambda_{\min} \cdot c_{\min} \cdot \|a\|_2^2 = \frac{m}{2} \cdot \|a\|_2^2$.

$\square$

**Lemma A.2.** *The gradient of the loss function $\nabla L_{ref}$ is $K$-Lipschitz with $K = \|X\|_{op}^2/4$.*

*Proof.* To show that $\nabla L_{\text{ref}}$ is $K$-Lipschitz, we need to show that $\nabla^2 L_{\text{ref}}(w) \preceq KI$. Recall from (6) that $\nabla^2 L_{\text{ref}}(w) = X^\top D(w)X$. Thus, we have

$$
\begin{aligned}
a^\top \nabla^2 L_{\text{ref}}(w)a &= a^\top X^\top D(w)Xa \\
&= \|D(w)^{1/2}Xa\|_2^2 \\
&\leq \|D(w)^{1/2}\|_{\text{op}}^2 \cdot \|X\|_{\text{op}}^2 \cdot \|a\|_2^2 \\
&\leq (\|X\|_{\text{op}}^2/4) \cdot \|a\|_2^2.
\end{aligned}
$$

In the final step, we use the fact that $D(w)$ is diagonal with non-zero elements in $(0, 1/4)$ to conclude that $\|D(w)^{1/2}\|_{\text{op}}^2 \leq 1/4$. $\square$

### A.1.2 CONVERGENCE OF GRADIENT DESCENT WITHIN THE REFERENCE SUBSPACE

Next, we establish that there exists a unique minimumizer of $L_{\text{ref}}$ within the reference subspace $W_{\text{ref}}$ (Lemma A.3) and that gradient descent converges to these weights (Lemma A.4).

**Lemma A.3.** *There exists a unique $w_{ref}^* \in W_{ref}$ such that $w_{ref}^* \in \arg\min_w L(w)$.*

*Proof.* We will first show that there exists a $w_{\text{ref}}^* \in W_{\text{ref}}$ such that $w_{\text{ref}}^* \in \arg\min_w L_{\text{ref}}(w)$. Let $w^* \in \arg\min_w L_{\text{ref}}(w)$ be an arbitrary minimimum point of $L_{\text{ref}}$. By definition, for every $(x, y) \in S_{\text{ref}}$, $x \in W_{\text{ref}}$. Hence, for every such $x$, $w^\top x = \text{proj}_{W_{\text{ref}}} w^\top x$. This means that $L_{\text{ref}}(w^*) = L_{\text{ref}}(\text{proj}_{W_{\text{ref}}} w^*)$, which implies that $w_{\text{ref}}^* := \text{proj}_{W_{\text{ref}}} w^* \in \arg\min_w L_{\text{ref}}(w)$, as desired. Next, because $L_{\text{ref}}$ is strictly convex on $W_{\text{ref}}$ (Lemma A.1), $w_{\text{ref}}^*$ is the only minimum point of $L_{\text{ref}}$ in $W_{\text{ref}}$. $\qquad\square$

**Lemma A.4.** *If we start with $w_{init} \in W_{ref}$ and run gradient descent with $\eta = 4/\|X\|_{op}^2$ to minimize $L_{ref}(w)$, the weights will converge to $w_{ref}^*$.*

*Proof.* Suppose that we start with initial weights $w_{\text{init}} \in W_{\text{ref}}$ and run gradient descent to minimize $L_{\text{ref}}$ with learning rate $\eta$. In particular, let $w^{(0)} = w_{\text{init}}$ and $w^{(t+1)} = w^{(t)} + \eta \cdot \nabla L_{\text{ref}}(w^{(t)})$. Because $L_{\text{ref}}$ is convex (Lemma A.1), $\nabla L_{\text{ref}}$ is $K$-Lipschitz with $K = \|X\|_{\text{op}}^2/4$ (Lemma A.2), and $\eta = 4/\|X\|_{\text{op}}^2 \leq 1/K$, we know from Theorem 3.2 of Bubeck (2014) that

$$L_{\text{ref}}(w^{(t)}) - L_{\text{ref}}(w_{\text{ref}}^*) \leq \frac{K \cdot \|w_{\text{init}} - w_{\text{ref}}^*\|}{t - 1}. \tag{8}$$

Hence, the loss attained by $w^{(t)}$ converges to the optimal loss attained by $w_{\text{ref}}^*$. To show that $w^{(t)}$ converges to $w_{\text{ref}}^*$, we will show that $L_{\text{ref}}$ is strongly convex on a set containing every $w^{(t)}$ for $t \geq 0$. In particular, consider the set $W_{\text{GD}} = \{w \in W_{\text{ref}} \mid \|w - w_{\text{ref}}^*\|_2 \leq \|w_{\text{init}} - w_{\text{ref}}^*\|_2\}$ containing weights in $W_{\text{ref}}$ at least as close to $w_{\text{ref}}^*$ as $w_{\text{init}}$. Clearly, $W_{\text{GD}}$ contains $w^{(0)} = w_{\text{init}}$. We know from Theorem 3.2 of Bubeck (2014) that with each iteration of gradient descent we get closer to a minimum point, that is, $\|w^{(t+1)} - w_{\text{ref}}^*\| \leq \|w^{(t)} - w_{\text{ref}}^*\|$. Additionally, because $w_{\text{init}}$ and $\nabla L_{\text{ref}}$ are in $W_{\text{ref}}$, every $w^{(t)}$ is in $W_{\text{ref}}$. Hence, every $w^{(t)}$ is in $W_{\text{GD}}$. Because $W_{\text{GD}}$ is closed and convex, from Lemma A.1 we know that $L_{\text{ref}}$ is strongly convex on $W_{\text{GD}}$. This means that there exists an $m > 0$ such that

$$L_{\text{ref}}(w^{(t)}) \geq L_{\text{ref}}(w_{\text{ref}}^*) + \nabla L_{\text{ref}}(w_{\text{ref}}^*)^\top (w^{(t)} - w_{\text{ref}}^*) + \frac{m}{2} \cdot \|w^{(t)} - w_{\text{ref}}^*\|_2^2.$$

Plugging in $\nabla L_{\text{ref}}(w_{\text{ref}}^*) = 0$ and rearranging, we get

$$\|w^{(t)} - w_{\text{ref}}^*\|_2^2 \leq \frac{2}{m} \cdot (L_{\text{ref}}(w^{(t)}) - L_{\text{ref}}(w_{\text{ref}}^*)).$$

Finally, combining with (8) yields

$$\|w^{(t)} - w_{\text{ref}}^*\|_2^2 \leq \frac{2 \cdot K \cdot \|w_{\text{init}} - w_{\text{ref}}^*\|}{m \cdot (t - 1)} \tag{9}$$

which completes our proof. $\qquad\square$

### A.1.3 PROOF OF THEOREM 3.1

We are now ready to prove Theorem 3.1. Suppose that we start with initial weights $w_{\text{init}}$ and run gradient descent to minimize $L_{\text{ref}}$ with learning rate $\eta = 4/\|X\|_{\text{op}}^2$. In particular, let $w^{(0)} = w_{\text{init}}$ and $w^{(t+1)} = w^{(t)} + \eta \cdot \nabla L_{\text{ref}}(w^{(t)})$ for $t \geq 0$. We will show that running gradient descent starting with an arbitrary $w_{\text{init}}$ has the same behavior as running gradient descent with $w_{\text{init}}$ projected onto $W_{\text{ref}}$. To be more precise, suppose that we instead start with initial weights $\text{proj}_{W_{\text{ref}}} w_{\text{init}}$ when running gradient descent. In particular, let $w_{\text{proj}}^{(0)} = \text{proj}_{W_{\text{ref}}} w_{\text{init}}$ and $w_{\text{proj}}^{(t+1)} = w_{\text{proj}}^{(t)} + \eta \cdot \nabla L_{\text{ref}}(w_{\text{proj}}^{(t)})$ for $t \geq 0$. Then the trajectory of $w^{(t)}$ is the same as that of $w_{\text{proj}}^{(t)}$ but with an additional component $(w_{\text{init}} - \text{proj}_{W_{\text{ref}}} w_{\text{init}})$, that is,

$$w^{(t)} = (w_{\text{init}} - \text{proj}_{W_{\text{ref}}} w_{\text{init}}) + w_{\text{proj}}^{(t)}.$$

To show that this is the case, we will proceed by induction. As a base case,

$$\begin{aligned} w^{(0)} &= w_{\text{init}} \\ &= w_{\text{init}} - \text{proj}_{W_{\text{ref}}} w_{\text{init}} + \text{proj}_{W_{\text{ref}}} w_{\text{init}} \\ &= (w_{\text{init}} - \text{proj}_{W_{\text{ref}}} w_{\text{init}}) + w_{\text{proj}}^{(0)}. \end{aligned}$$

For the inductive step, assume that the statement holds for $t = k$. Then,

$$
\begin{aligned}
w^{(k+1)} &= w^{(k)} + \eta \cdot \nabla L_{\text{ref}}(w^{(k)}) \\
&= (w_{\text{init}} - \text{proj}_{W_{\text{ref}}} w_{\text{init}}) + w_{\text{proj}}{}^{(k)} + \eta \cdot \nabla L_{\text{ref}}((w_{\text{init}} - \text{proj}_{W_{\text{ref}}} w_{\text{init}}) + w_{\text{proj}}{}^{(k)}) \\
&= (w_{\text{init}} - \text{proj}_{W_{\text{ref}}} w_{\text{init}}) + w_{\text{proj}}{}^{(k)} + \eta \cdot \nabla L_{\text{ref}}(w_{\text{proj}}{}^{(k)}) \\
&= (w_{\text{init}} - \text{proj}_{W_{\text{ref}}} w_{\text{init}}) + w_{\text{proj}}{}^{(k+1)}
\end{aligned}
$$

where in the third step we use the fact that $\nabla L_{\text{ref}}(w) = \nabla L_{\text{ref}}(\text{proj}_{W_{\text{ref}}} w)$. This completes the induction. Because $w_{\text{proj}}{}^{(0)} = \text{proj}_{W_{\text{ref}}} w_{\text{init}} \in W_{\text{ref}}$, from Lemma A.4 (in particular, from equation 9), we know that

$$
\|w_{\text{proj}}{}^{(t)} - w_{\text{ref}}^*\|_2^2 \leq \frac{2 \cdot K \cdot \|\text{proj}_{W_{\text{ref}}} w_{\text{init}} - w_{\text{ref}}^*\|}{m \cdot (t-1)}.
$$

where $K$ and $m$ are positive constants. Finally, we conclude that

$$
\begin{aligned}
\|w^{(t)} - \hat{w}\|_2^2 &= \|((w_{\text{init}} - \text{proj}_{W_{\text{ref}}} w_{\text{init}}) + w_{\text{proj}}{}^{(t)}) - ((w_{\text{init}} - \text{proj}_{W_{\text{ref}}} w_{\text{init}}) + w_{\text{ref}}^*)\|_2^2 \\
&= \|w_{\text{proj}}{}^{(t)} - w_{\text{ref}}^*\|_2^2 \\
&\leq \frac{2 \cdot K \cdot \|\text{proj}_{W_{\text{ref}}} w_{\text{init}} - w_{\text{ref}}^*\|}{m \cdot (t-1)}
\end{aligned}
$$

Hence, $w^{(t)}$ converges to $\hat{w}$, completing our proof.

# B EXPERIMENT DETAILS

## B.1 GENERAL

### B.1.1 MODEL TRAINING

All models are trained using the FFCV data-loading library (Leclerc et al., 2022) on a cluster of A100 GPUs.

### B.1.2 MEASURING EFFECTIVE ROBUSTNESS

**Effective robustness.** In this work, we quantify the robustness of pre-trained models using *effective robustness* (ER), a measure of the robustness a model above the "baseline" of models trained from scratch (Taori et al., 2020). Computing this metric first involves establishing a relationship between the accuracies of baseline models (in our case, models trained from scratch on a reference dataset). In particular, let $\text{Acc}_{\text{ref}}(M)$ and $\text{Acc}_{\text{shift}}(M)$ denote the accuracies of a model $M$ on test datasets drawn from the reference and shifted distributions, respectively. Given a set $\mathcal{M}_{\text{baseline}}$ of baseline models, we compute a linear fit relating $\Phi^{-1}(\text{Acc}_{\text{ref}}(M))$ and $\Phi^{-1}(\text{Acc}_{\text{shift}}(M))$, where $\Phi^{-1}$ is the probit function (i.e., the inverse cumulative distribution function of the standard normal distribution). We compute a linear fit relating probit-scaled accuracies because this has been empirically observed to improve the strength of the linear relationship (Miller et al., 2021; Taori et al., 2020). Formally, we compute parameters $\hat{a}$ and $\hat{b}$ such that

$$\hat{a}, \hat{b} = \arg\min_{a,b} \sum_{M \in \mathcal{M}_{\text{baseline}}} \|(a \cdot \Phi^{-1}(\text{Acc}_{\text{ref}}(M)) + b) - \Phi^{-1}(\text{Acc}_{\text{shift}}(M))\|_2.$$

Let $\widehat{\text{Acc}}_{\text{shift}}(M)$ be the resulting function estimating shifted accuracy given reference accuracy, that is

$$\widehat{\text{Acc}}_{\text{shift}}(M) = \Phi(\hat{a} \cdot \Phi^{-1}(\text{Acc}_{\text{ref}}(M)) + \hat{b}).$$

Then the effective robustness of a model $M$ is

$$\text{ER}(M) = \widehat{\text{Acc}}_{\text{shift}}(M) - \text{Acc}_{\text{shift}}(M)$$

Intuitively, effective robustness is the extent to which a model's accuracy on the shifted distribution exceeds the accuracy of a baseline model with the same accuracy on the reference distribution.

**Establishing a baseline for effective robustness.** To establish a baseline with respect to which we can measure effective robustness, we train ResNet-18 models from scratch on differently sized subsets of the reference dataset. The number of models and the minimum subset size vary by experiment. Miller et al. (2021) observe that models trained from scratch in this way often exhibit a strong linear relationship between their accuracies on the reference and shifted distributions (and the same relationship holds for models with different architectures, hyperparameters, etc.). In each of the experiments in which we measure effective robustness, we confirm that this relationship exists for our baseline models (see, e.g., Figure 3).

## B.2 CONSTRUCTING SYNTHETIC IN-SUPPORT AND OUT-OF-SUPPORT SHIFTS

In Section 4.1, we measure the effective robustness of various pre-trained and fine-tuned models on two in-support and two out-of-support shifts synthetically constructed by modifying CIFAR-10.

**Specifications of synthetic shifts.** Here, we provide detailed descriptions of the four synthetic distribution shifts (see Figure 3 for visualizations).

1. **Tint shift** (in-support): We tint images (i.e., replace each pixel with a mix of the original value, with weight $0.75$ and a specific color, with weight $0.25$) such that the tint is correlated with the label in the reference distribution but not in the shifted distribution (i.e., tint is a spurious feature). Specifically, in the reference distribution we apply tint with a class-specific color to 80% of examples and a tint with a random color to the remaining 20%. Meanwhile, in the shifted distribution we apply a tint with a random color universally.

2. **Label shift** (in-support): Label shift is a commonly studied type of distribution shift in which the relative frequencies of classes change, but $p(x|y)$ is fixed. To construct a label shift, we sub-sample CIFAR-10 such that in the reference distribution, the first five classes are four times more likely to appear than the last five classes. In the shifted distribution, these relative frequencies are reversed.

3. **Pad shift** (out-of-support): We pad the images in the shifted distribution by adding 6 black pixels to each side of the original $32 \times 32$ CIFAR-10 images. Note that our models resize inputs to $224 \times 224$ as a pre-processing step, so padding does not affect the final size of images fed into our models.

4. **Flip shift** (out-of-support): We vertically flip images in the shifted distribution.

**Shared model specifications.** For data augmentation, we use the FFCV implementations of *RandomHorizontalFlip* and *RandomTranslate*. We preprocess images by resizing the original $32 \times 32$ images to a resolution of $224 \times 224$.

**Specifications of baseline models.** To establish a baseline, we train $64$ models from scratch on subsets ranging from 50% of the dataset to the entire dataset. We train baseline models by running SGD for 96 epochs, using a triangular learning rate schedule with a peak learning rate of $0.5$ and $5$ warmup epochs, a batch size of $512$, a weight decay of $5 \times 10^{-4}$ and a momentum of $0.9$.

**Specifications of pre-trained models and fine-tuning strategies.** We consider a 7 different pre-trained models (implementations from PyTorch Image Models (Wightman, 2019)): ResNet-18 and ResNet-50 models trained on ImageNet-1K (Deng et al., 2009), a Big Transfer (BiT) (Kolesnikov et al., 2019) ResNet-50[3] model trained on ImageNet-21K, a BEiT (Bao et al., 2021) model with patch size of $16 \times 16$ and resolution of $224 \times 224$ trained on ImageNet-22K (with self-supervised training followed by standard supervised training), and CLIP models from Radford et al. (2021) with ResNet-50, ViT-B/32 and ViT-B-16 architectures. For the ImageNet-1K models, BiT and BEiT models, we employ full fine-tuning with a randomly initialized classification layer. For the CLIP models, in addition to full fine-tuning we also consider linear probing followed by full fine-tuning (LP-FT) (Kumar et al., 2022) and full fine-tuning initialized with zero-shot weights, as specified by Radford et al. (2021). This results in a total of 13 pre-trained and fine-tuned models, which we group in Figure 3 according to the pre-training and fine-tuning strategies (models within a single group have different architectures, but are otherwise the same).

We fine-tune models by running Adam for $24$ epochs, using a cosine learning rate schedule with $5$ warmup epochs. We select the best peak learning rate (in terms of reference accuracy) among $1 \times 10^{-2}, 3 \times 10^{-3}, 1 \times 10^{-3}, 3 \times 10^{-4}, 1 \times 10^{-4}, 3 \times 10^{-5}, 1 \times 10^{-5}, 3 \times 10^{-6}$. We use a batch size of $256$ for the BiT, BEiT and ViT-B/16 models (due to memory constraints) and a batch size of $512$ for other pre-trained models and use a weight decay of $0.1$. When doing LP-FT, we perform linear probing with the same hyperparameters as full fine-tuning, except that we always use a learning rate of $0.1$ and a weight decay of $0$.

### B.3 DIVIDING NATURAL SHIFTS INTO IN-SUPPORT AND OUT-OF-SUPPORT SPLITS

#### B.3.1 SPLITTING A SHIFTED DATASET

To split a shifted dataset into an "in-support split" and an "out-of-support split", we would ideally measure the reference distribution probability density $p_{\text{ref}}$ of inputs in the shifted dataset and assign inputs with small $p_{\text{ref}}$ to the out-of-support split. Unfortunately, it is difficult to estimate $p_{\text{ref}}$ directly when dealing with high-dimensional inputs (in this case, images). Instead, we estimate the probability density *ratio* $p_{\text{ref}}/p_{\text{shift}}$, that is, how much more likely an input is under the reference distribution than under the shifted distribution. We then assign examples in the shifted dataset with $p_{\text{ref}}/p_{\text{shift}} < 0.2$ to the out-of-support split and examples with $p_{\text{ref}}/p_{\text{shift}} \geq 0.2$ to the in-support split. We visualize examples in Figure 11.

**Estimating $p_{\text{ref}}/p_{\text{shift}}$.** To estimate $p_{\text{ref}}/p_{\text{shift}}$, we use a classifier trained to distinguish between examples from the reference and shifted datasets. Specifically, let $p$ be a probability mass/density function over examples that can either be drawn from $\mathcal{D}_{\text{ref}}$ or $\mathcal{D}_{\text{shift}}$ (i.e., $p$ represents the distribution

---

[3]We use the ResNet-v2 variant of the ResNet-50 architecture from He et al. (2016).

of a dataset created by joining a reference dataset and a shifted dataset). Next, let $y_{\text{ref}}$ be the event that an example is drawn from $\mathcal{D}_{\text{ref}}$ and $y_{\text{shift}}$ be the event that an example is drawn from $\mathcal{D}_{\text{shift}}$. We can express the ratio $p_{\text{ref}}/p_{\text{shift}}$ as follows:

$$\frac{p_{\text{ref}}(x)}{p_{\text{shift}}(x)} = \frac{p(x|y_{\text{ref}})}{p(x|y_{\text{shift}})}$$

$$= \frac{p(y_{\text{ref}}|x) \cdot p(x)}{p(y_{\text{ref}})} \cdot \frac{p(y_{\text{shift}})}{p(y_{\text{shift}}|x) \cdot p(x)}$$

$$= \frac{p(y_{\text{ref}}|x)}{p(y_{\text{shift}}|x)} \cdot \frac{p(y_{\text{shift}})}{p(y_{\text{ref}})}.$$

The terms $p(y_{\text{ref}})$ and $p(y_{\text{shift}})$ are easy to estimate since they are simply the proportions of reference and shifted examples in $p$. Hence, to estimate $p_{\text{ref}}/p_{\text{shift}}$ we just need to estimate $p(y_{\text{ref}}|x)$ and $p(y_{\text{shift}}|x)$.

To do so, we train a classifier to distinguish between reference and shifted examples on a dataset drawn from $p$. We construct such a dataset by combining $100K$ samples from ImageNet with each of the shifted datasets (for ImageNet-R, which contains a subset of the classes of ImageNet, we restrict the $100K$ samples to these classes). Next, we fine-tune a CLIP ViT-L/14 pre-trained on LAION-2B from OpenCLIP (Ilharco et al., 2021) to distinguish between reference and shifted examples. We first fine-tune just the final layer with a learning rate of $0.1$ and then fine-tune the entire model with the best learning rate selected from $2 \times 10^{-4}, 1 \times 10^{-4}, 5 \times 10^{-5}, 2 \times 10^{-5}, 1 \times 10^{-5}, 5 \times 10^{-6}, 2 \times 10^{-6}$ and $1 \times 10^{-6}$. After training the classifier, we calibrate it by rescaling its output. We then estimate $p(y_{\text{ref}}|x)$ and $p(y_{\text{shift}}|x)$ by applying a sigmoid to its output, from which we can estimate $p_{\text{ref}}/p_{\text{shift}}$. To estimate this ratio for the entire shifted dataset, we split the dataset into 10 folds and train a classifier to estimate $p_{\text{ref}}/p_{\text{shift}}$ on each fold using the remaining 9 folds.

**Calibrating the classifiers used for splitting** As discussed in Section B.3, our method for dividing a shifted dataset into an in-support split and an out-of-support split requires a *calibrated* classifier to distinguish between examples from the reference and shifted datasets. Recall that to distinguish between examples from the reference and shifted datasets, we fine-tune a CLIP ViT-L/14 pre-trained on LAION-2B from OpenCLIP. Such over-parameterized models can be overconfident in their predictions (and thus uncalibrated), so we calibrate the classifier by rescaling its (logit) output.

In particular, let $f$ be a (potentially uncalibrated) classifier trained to distinguish between examples from the reference and shifted datasets (where the output of $f$ is a logit). We find the scaling parameter $\alpha$ that minimizes the standard logistic loss of $f$ on a calibration set $S_{\text{cal}}$:

$$\alpha = \arg\min_{\alpha'} \sum_{(x,y) \in S_{\text{cal}}} \log(1 + e^{-\alpha' \cdot f(x) \cdot y}). \tag{10}$$

We then define a rescaled classifier $f_{\text{cal}}(x) = \alpha \cdot f(x)$ (which is used to estimate the ratio $p_{\text{ref}}/p_{\text{shift}}$). We produce calibration curves of the rescaled classifiers for each of the shifted datasets we split (see Figure 7) and observe that they are indeed well-calibrated.

### B.3.2 SPECIFICATIONS OF IMAGENET MODELS

To measure the robustness benefits of pre-training on in-support and out-of-support splits of ImageNet distribution shifts, we take existing models from PyTorch Image Models (Wightman, 2019). We establish a baseline for robustness by taking 77 models of different architectures trained from scratch on ImageNet. We compute the effective robustness of 24 pre-trained models that are fine-tuned on ImageNet. We select only models pre-trained on large web-scale datasets such as LAION-2B Schuhmann et al. (2022) and IG-1B (Mahajan et al., 2018), as these models are likely to have higher effective robustness.

### B.4 COMBINING PRE-TRAINING WITH INTERVENTIONS FOR HANDLING BIAS

**Shared model specifications.** When training on the WILDS-FMoW dataset, we use the FFCV implementation of *RandomHorizontalFlip*. We train on a randomly sampled subset of the training dataset that is $50\%$ of the size of the original training dataset and reserve the remaining $50\%$ for the *Deep Feature Reweighting* intervention.

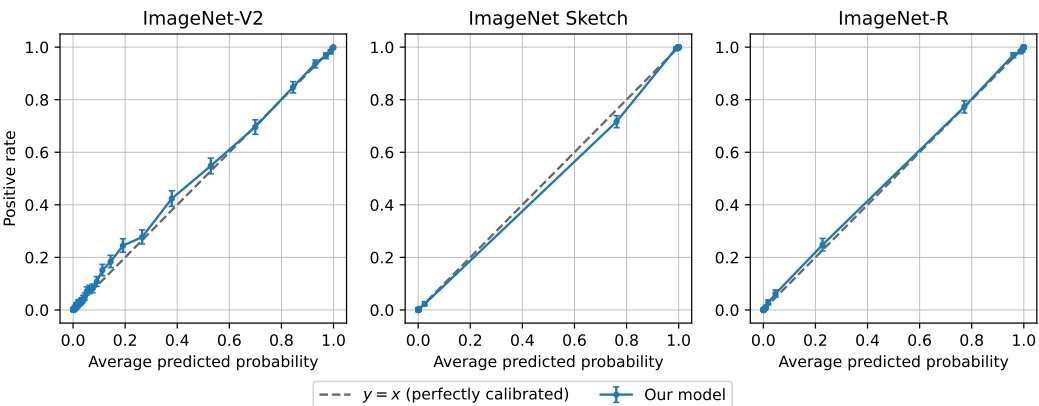

Figure 7: **Calibration curves of classifiers used for splitting.** We display calibration curves for the classifiers used to divide ImageNet-V2, ImageNet-Sketch and ImageNet-R into in-support and out-of-support splits. Specifically, we sort the outputs of each classifier on a combined dataset of reference and shifted examples into 100 bins (where bin edges are quantiles). For each bin, we compute the actual positive rate (i.e., the proportion of examples from the shifted dataset) and the average predicted probability of an example being from the shifted dataset. When we plot the actual positive rates against average predicted probabilities, they are close to equal (close to $y = x$), suggesting that the classifiers are well-calibrated. Error bars denote 95% Clopper-Pearson confidence intervals.

**Specifications of models trained from scratch.** We train models from scratch by running SGD for 16 epochs, using a triangular learning rate schedule with a peak learning rate of $0.5$ and 2 warmup epochs, a batch size of $128$, a weight decay of $5 \times 10^{-4}$ and a momentum of $0.9$.

**Baseline specifications.** To establish a baseline, we train $64$ models from scratch on subsets ranging from 25% of the dataset to the entire dataset.

**Specifications of pre-trained models.** The pre-trained model in this experiment is a CLIP ViT-B/32 model initialized as a zero-shot classifier following Wortsman et al. (2021). We fine-tune models by running AdamW for $10$ epochs, using a cosine learning rate schedule with a peak learning rate of $3 \times 10^{-5}$ and 2 warmup epochs, a batch size of $512$ and a weight decay of $0.1$.

**Our implementation of _Deep Feature Reweighting_.** The _Deep Feature Reweighting_ (DFR) intervention proposed by Kirichenko et al. (2022) aims to improve the robustness of a model on difficult subpopulations by using a validation dataset with group labels. The algorithm consists of two steps: (1) train a standard model on the original training dataset, and (2) re-train only the final layer of the model (i.e., "re-weight" the features of the model) on the validation dataset to be more favorable to minority groups. To re-train the final layer, Kirichenko et al. (2022) repeatedly sample group-balanced subsets of the validation dataset, re-train the final layer on each subset, and then average the resulting re-trained final layers. Our implementation differs slightly in that we assign sample weights to the validation dataset such that each group has equal total weight and re-train the final layer on the weighted validation dataset. When applying _Deep Feature Reweighting_ to WILDS-FMoW, we use 50% of the training dataset for re-weighting.

### B.5 CURATING DATASETS FOR FINE-TUNING

**Image editing to synthesize "counterfactual examples"** In order to curate a "de-biased" dataset for hair color classification, we edit images from CelebA-HQ (Karras et al., 2018), a subset of the CelebA dataset with segmentation masks for each attribute provided by CelebAMask-HQ (Lee et al., 2020). To change the hair color in a given image, we use InstructPix2Pix (Brooks et al., 2023), a recent image editing model fine-tuned from Stable Diffusion (Rombach et al., 2022). This model accepts an input image to be edited along with a prompt describing the desired change (e.g., "change the hair color to blond"). We find that InstructPix2Pix is able to successfully edit the hair color; however, this model often makes undesired changes to attributes such as skin tone and eye color

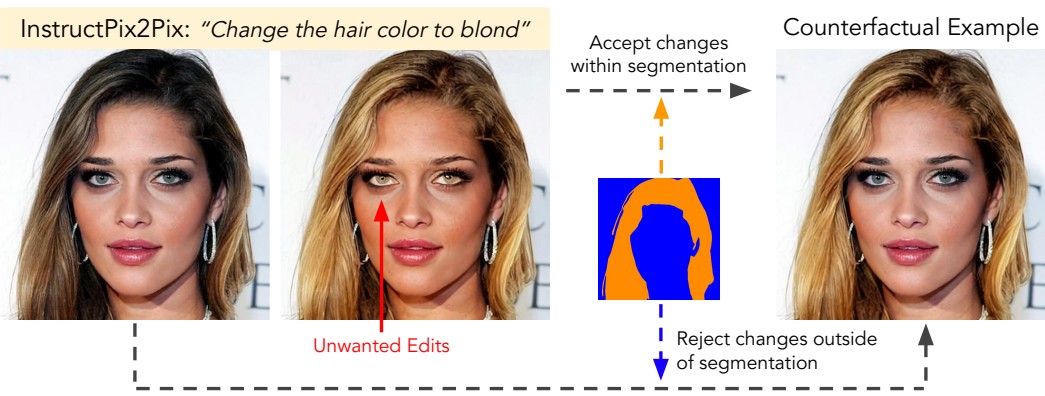

Figure 8: **Synthesizing counterfactual examples.** We edit hair color in CelebA-HQ images using InstructPix2Pix (Brooks et al., 2023). However, this model can also make unwanted changes to attribute other than hair color, e.g., changing eye color (left). To avoid such issues, in the final image we incorporate only changes within the hair region of the image.

(see, e.g., the left side of Figure 8). To ensure that we only edit hair color, we use the attribute masks to isolate the pixels in a given image corresponding to the hair region, and ignore any changes made outside of this area. When using a binary mask, this procedure could cause unnatural "edges" along the border of the mask. Thus, we apply a Gaussian blur to the hair mask to smooth the transition when "merging" the original and edited images.

To edit an image from non-blond to blond, we use the prompt "change the hair color to blond." When editing from blond to non-blond, however, we find that the prompt "change the hair color to non-blond" gives inconsistent results, likely because the instruction is vague. We observe that most non-blond people in the CelebA dataset have brown or black hair, so as a simple heuristic we randomly edit each image with either the prompt "change the hair color to brown" or the prompt "change the hair color to black." See Figure 8 for a visualization of the image editing process.

**Shared model specifications.** Accuracy and worst-group accuracy on the CelebA dataset are sensitive to hyperparameter choices. As a result, we conduct a grid search to select hyperparameters for each type of model. We use class-balanced accuracy as the metric for hyperparameter selection, which empirically better correlates with worst-group accuracy than standard accuracy.

When selecting hyperparameters for a curated dataset of a given size, we randomly sample $32$ datasets of that size from a pool of $16,000$ images (i.e., $8,000$ CelebA images and their corresponding counterfactual synthesized images) and average the class-balanced accuracies of models trained on each dataset. When evaluating the accuracy and worst-group accuracy of models trained on a curated dataset of a given size, we similarly randomly sample $64$ datasets of that size and report average metrics.

For all models, we use the FFCV implementation of *RandomHorizontalFlip* for data augmentation.

**Specifications of models trained from scratch.** We train ResNet-18 models from scratch by running SGD for 32 epochs, using a triangular learning rate schedule with $4$ warmup epochs. We use a batch size of 128, a weight decay of $5 \times 10^{-4}$ and a momentum of $0.9$. We select the best combination of batch size and learning rate from batch sizes of $64, 128, 256, 512$ and learning rates of $0.5, 0.2, 0.1, 0.05, 0.02, 0.01$.

When training models from scratch on our curated dataset, we run SGD for $512$ epochs and use a triangular learning rate schedule with $64$ warmup epochs. We use a batch size equal to the total number of examples when it is less than $512$ and a batch size of $512$ otherwise. We use a weight decay of $5 \times 10^{-4}$ and a momentum of $0.9$. We select the best learning rate from $0.5, 0.2, 0.1, 0.05, 0.02, 0.01$.

**Baseline specifications.** To establish a baseline, we train 100 models from scratch on subsets ranging from 5% of the dataset to the entire dataset.

**Specifications of pre-trained models.** The pre-trained model in this experiment is a CLIP ViT-B/32 model initialized as a zero-shot classifier with "blond" and "non-blond" as the class names. We fine-tune models by running AdamW for 16 epochs, using a cosine learning rate schedule with 2 warmup epochs, and a weight decay of $0.1$. We select the best combination of batch size and learning rate from batch sizes of $64, 128, 256, 512$ and learning rates of $3\times10^{-5}, 1\times10^{-5}, 3\times10^{-6}, 1\times10^{-6}$.

When training on our curated dataset, we use a batch size of $64$ (the size of the dataset) and select the best learning rate from $3 \times 10^{-5}, 1 \times 10^{-5}, 3 \times 10^{-6}, 1 \times 10^{-6}$.

# C    ADDITIONAL RESULTS

## C.1    CONSTRUCTING SYNTHETIC IN-SUPPORT AND OUT-OF-SUPPORT SHIFTS

### C.1.1    HOW DOES THE CHOICE OF FINE-TUNING HYPERPARAMETERS AFFECT ROBUSTNESS?

In Section 4.1, we select hyperparameters (in particular, learning rate) for fine-tuning that maximize accuracy on the reference distribution. This reasonably simulates hyperparameter selection in practice because typically only samples from the reference distribution are available.

In this section, we investigate how the choice of hyperparameters affects the robustness of pre-trained models. In particular, we would like to understand if pre-training yields little effective robustness to in-support shifts and substantial effective robustness to out-of-support shifts across a wider range of hyperparameter choices. We study the tint shift (an in-support shift) and the pad shift (an out-of-support shift) from Section 4.1 and vary the learning rate, weight decay, number of epochs, and batch size of a CLIP ViT-B/32 initialized with zero-shot weights (Figure 9). With zero-shot initialization, the starting point of fine-tuning is a robust model that performs well on our task. Hence, even under an in-support shift, hyperparameter choices that do not change the model substantially (e.g., low learning rate, small number of epochs) result in substantial effective robustness. However, these hyperparameter choices generally result in lower absolute reference and shifted accuracies, and are thus unreasonable. The hyperparameter choices that are relevant in practice are those with high reference accuracy, and these are the hyperparameters that we use in our experiments.

### C.1.2    WHY DO PRE-TRAINED MODELS EXHIBIT ANY EFFECTIVE ROBUSTNESS UNDER IN-SUPPORT SHIFTS?

On the left side of Figure 3, we observe a small robustness gap between pre-trained models and models trained from scratch on in-support shifts constructed with CIFAR-10. This seemingly refutes our intuition that pre-training helps specifically with extrapolation, which implies that in-support, pre-training should offer *no* robustness benefits. However, we hypothesize that this robustness gap actually stems from extrapolation: under empirical shifts (where we have access to samples and not an actual distribution), models must extrapolate from the limited number of samples available. For example, in the label shift setting there are only $1,250$ instances of each of the minority classes in the training dataset, and these samples may not fully represent the distributions of these classes.

To determine whether the limited sample size explains the robustness gap, we repeat our experiment from Section 4.1 on 1 million examples (a $20\times$ increase in size) from the CIFAR-5m dataset (Nakkiran et al., 2020), a large synthetically generated dataset resembling CIFAR-10. Besides the number of epochs, which we decrease due to the larger dataset size, we use the same hyperparameters as in the original experiment and again select the learning rate for fine-tuning that maximizes reference accuracy. We observe that with a larger sample size, the effective robustness of pre-trained models on in-support shifts vanish (Figure 10). This suggests that under the CIFAR-10 in-support shifts, the small effective robustnesses of pre-trained models are due to extrapolation from the small number of samples available, which would be consistent with our hypothesis. We also observe decreases in the effective robustnesses of pre-trained models on out-of-support shifts, though pre-trained models still exhibit substantial robustness under these shifts. This may be because as the size of the fine-tuning dataset increases, the pre-trained features change more during fine-tuning, leading to lower robustness.

## C.2    DIVIDING NATURAL SHIFTS INTO IN-SUPPORT AND OUT-OF-SUPPORT SPLITS

### C.2.1    SIZES OF IN-SUPPORT AND OUT-OF-SUPPORT SPLITS

In Table 1, we report the sizes of the in-support and out-of-support splits we compute for ImageNet-V2, ImageNet Sketch and ImageNet-R. The out-of-support splits are much larger than the in-support splits, perhaps because the large majority of the examples from these shifted datasets look unlike examples from ImageNet.

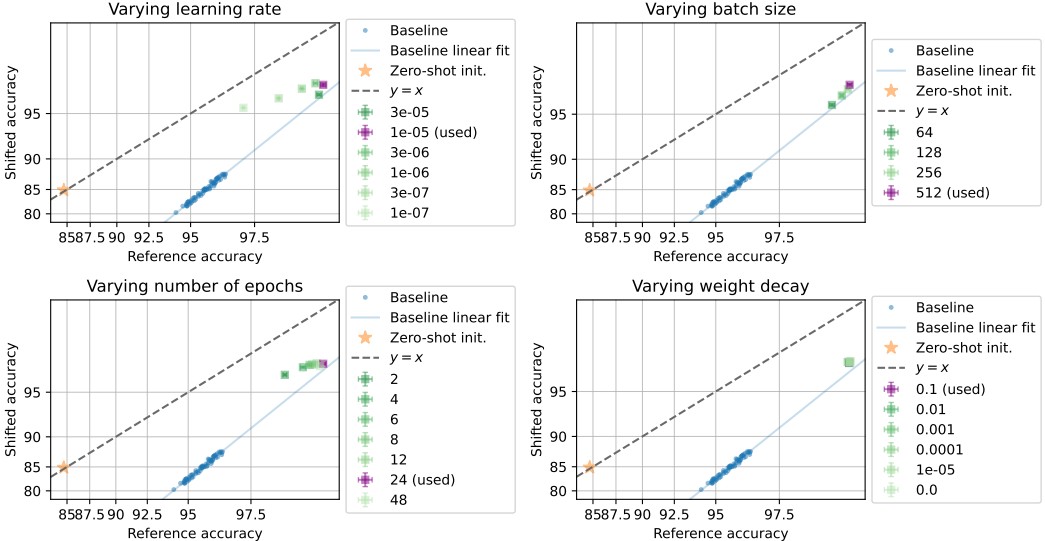

(a) **In-support shift.** The in-support shift we consider is the "tint shift" in which we introduce a tint that is spuriously correlated with the label. On this in-support shift, learning rate and number of epochs influence effective robustness, but the best hyperparameter choices result in a model with little effective robustness.

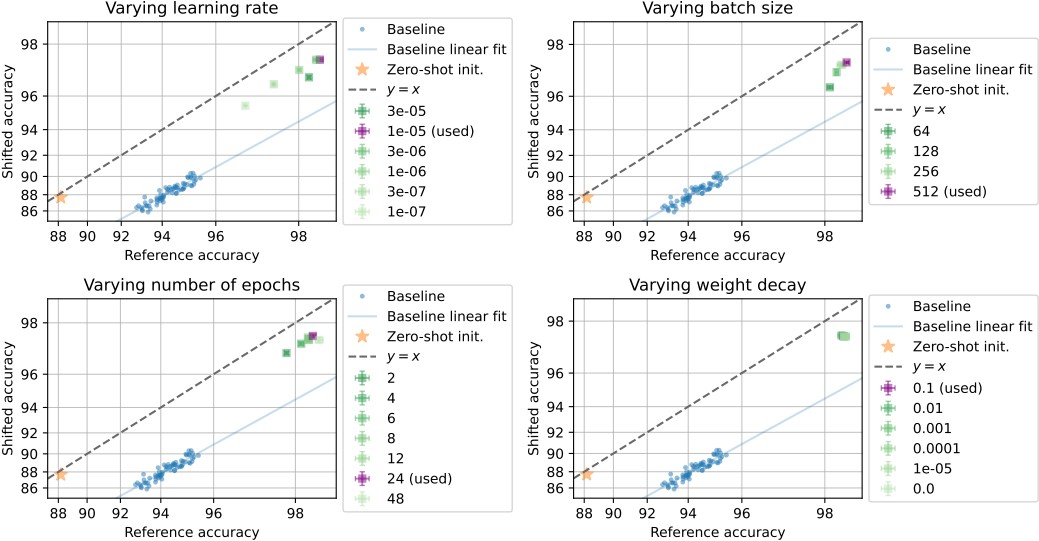

(b) **Out-of-support shift.** The out-of-support shift we consider is the "pad shift" in which we pad images in the shifted distribution. On this out-of-support shift, batch size most significantly affects robustness, while learning rate and number of epochs affect overall performance.

Figure 9: **The effects of hyperparameter choices on robustness.** We vary hyperparameters when fine-tuning a CLIP ViT-B/32 initialized with zero-shot weights on synthetic CIFAR-10 shifts from Section 4.1 (different shades of green). Varying certain hyperparameters (e.g., learning rate, number of epochs) can affect the effective robustness of pre-trained models even on an in-support shift. In our experiments, we choose hyperparameters which yield high reference accuracy (purple).

### C.2.2 EXAMPLES FROM IN-SUPPORT AND OUT-OF-SUPPORT SPLITS

In Figure 11, we provide samples from the in-support and out-of-support splits we compute for ImageNet-V2, ImageNet-Sketch and ImageNet-R.

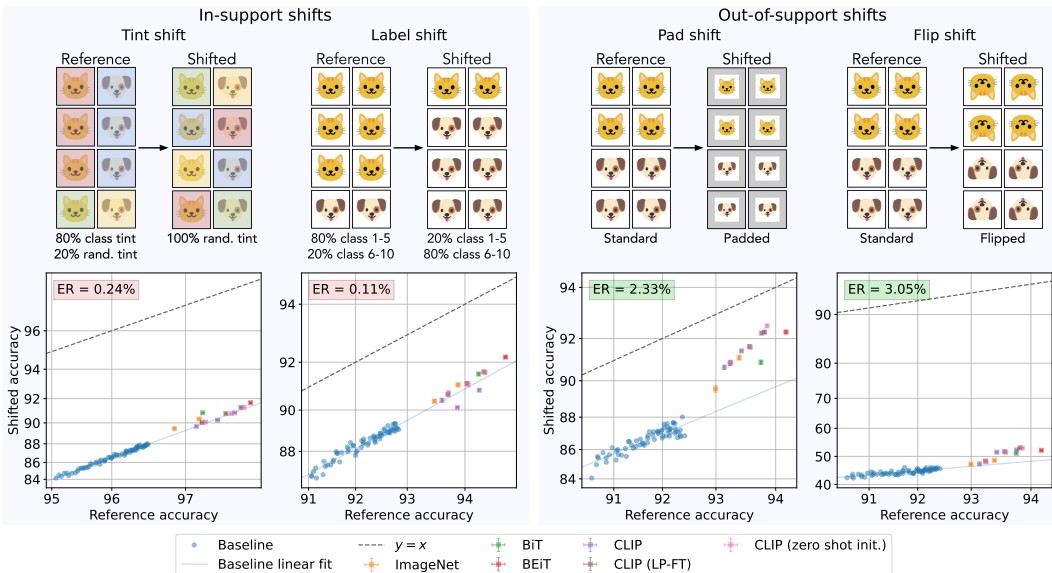

Figure 10: **Robustness of pre-trained models to synthetic in-support and out-of-support shifts constructed from CIFAR-5m.** Pre-trained models exhibit virtually no effective robustness on the in-support shifts (left), but have significant effective robustness on the out-of-support shifts (right). Error bars denote 95% confidence intervals over 8 random trials.

Table 1: Sizes of in-support and out-of-support splits.

| Dataset | In-support split size | Out-of-support split size |
|---|---|---|
| ImageNet-V2 | 1920 | 8080 |
| ImageNet Sketch | 162 | 50727 |
| ImageNet-R | 588 | 29412 |

### C.2.3 SCATTER PLOTS OF REFERENCE VS. SHIFTED ACCURACY

In Figure 12, we provide scatter plots of accuracy on ImageNet vs. accuracy on the in-support and out-of-support splits of ImageNet-V2, ImageNet Sketch and ImageNet-R.

### C.2.4 CONTROLLING FOR DIFFICULTY WHEN MEASURING EFFECTIVE ROBUSTNESS

The significance of a given effective robustness depends on the "difficulty" of a distribution shift. For example, if a shift causes an accuracy drop of 5%, an effective robustness of 4% might be considered large, but if a shift that causes a drop of 25%, an effective robustness of 4% would probably be considered small. When we divide a shifted dataset into an in-support and out-of-support split, the out-of-support split is typically more difficult than the in-support split. If we compare the effective robustness of pre-trained models on examples of similar difficulty in the in-support and out-of-support splits, do our findings from Section 4.2 still hold? In particular, do pre-trained models still exhibit substantially higher robustness on out-of-support examples than on in-support examples?

To answer this question, we re-weight examples in out-of-support splits such that the difficulty distribution of the out-of-support split matches that of the in-support split. Specifically, we quantify the difficulty of a given example in terms of the fraction of baseline models (of 77 total baseline models) that classify it incorrectly. Given an example of difficulty $d$, we re-weight it by a factor of $p_{\text{in-support}}(d)/p_{\text{out-of-support}}(d)$ where $p_{\text{in-support}}$ is the difficulty probability density function of the in-support split and $p_{\text{out-of-support}}$ is the difficulty probability density function of the out-of-support split. We then compute a "re-weighted" accuracy, which in turn yields a re-weighted effective robustness, on the out-of-support split. Intuitively, this re-weighted effective robustness represents

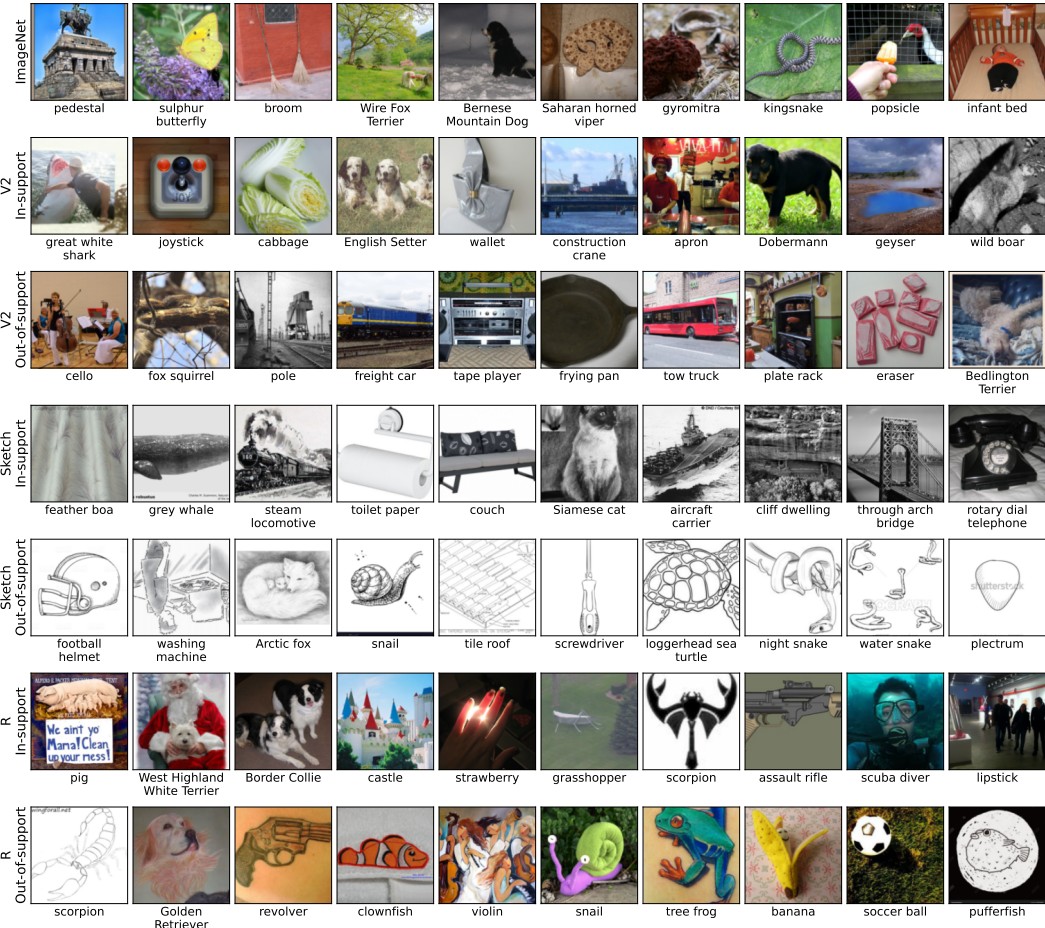

Figure 11: **Random samples from ImageNet and from the in-support and out-of-support splits of ImageNet-V2, ImageNet Sketch and ImageNet-R.** In ImageNet-V2, it is difficult to distinguish between examples from the in-support and out-of-support splits. In ImageNet Sketch and ImageNet-R, examples from the in-support splits look more realistic (i.e., more like ImageNet examples) than examples from the out-of-support splits.

the effective robustness of pre-trained models on out-of-support examples of similar difficulty to in-support examples.

We report the re-weighted effective robustnesses in Figure 13. We observe that the re-weighted effective robustnesses of pre-trained models on out-of-support splits are indeed lower than the original effective robustnesses. However, they are still substantially higher than the effective robustnesses on in-support splits.

### C.3  COMBINING PRE-TRAINING WITH INTERVENTIONS FOR HANDLING BIAS

#### C.3.1  STUDYING A SYNTHETIC SHIFT

In this section, we provide an additional experiment in a synthetic setting to further illustrate that pre-training and interventions designed to handle dataset biases can be complementary. In Section 5, we discussed how robustness to the WILDS-FMoW distribution shift requires both extrapolating to later years and performing consistently across regions. We construct a synthetic distribution shift using that similarly requires both extrapolating well and avoiding reliance on spurious features. Specifically, we combine the tint and pad shifts from Section 4.1. We modify CIFAR-10 such that in the reference distribution, we add a tint that is spuriously correlated with the label: $80\%$ of reference

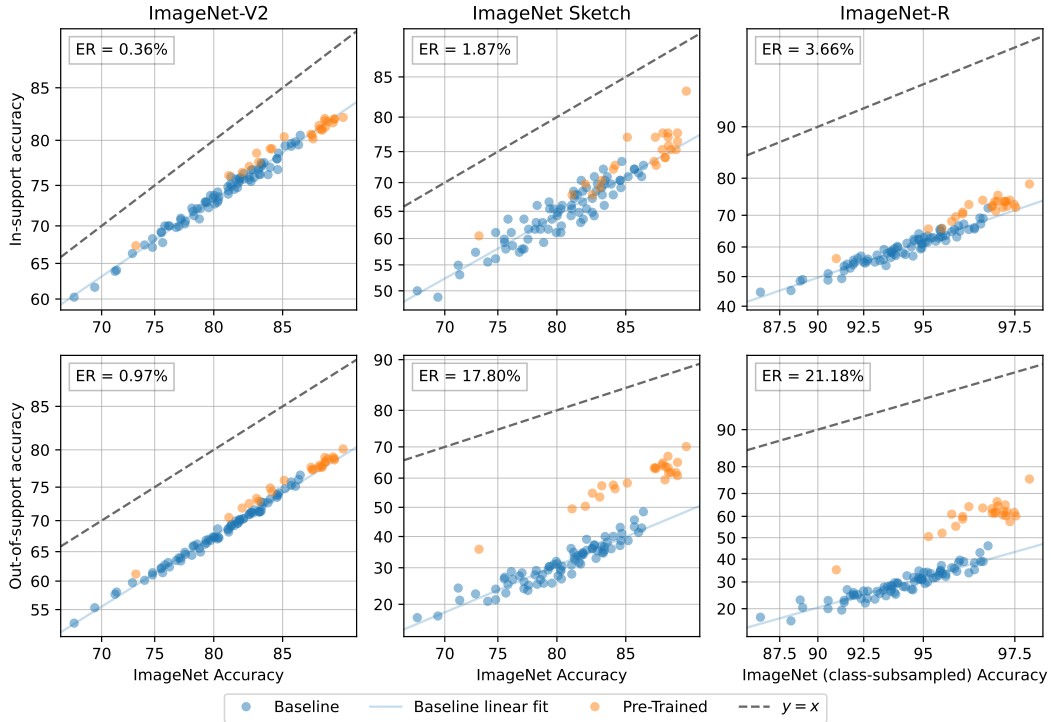

Figure 12: **Reference vs. shifted accuracy for in-support and out-of-support splits of ImageNet shifts.** On each of the three ImageNet shifts we consider, the average effective robustness (ER) of pre-trained models (orange) above the baseline of models trained from scratch (blue) on the in-support split (top) is small. Meanwhile, their effective robustness can be very large on the out-of-support split (bottom).

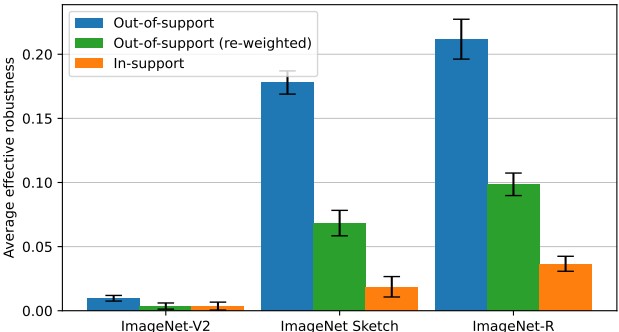

Figure 13: **Re-weighed effective robustness of pre-trained models on in-support and out-of-support splits of ImageNet shifts.** When we re-weight examples in out-of-support splits to match the difficulty distributions of their corresponding in-support splits, the average effective robustnesses of pre-trained models (green) decrease relative to the original effective robustnesses (blue). However, they are still very high on ImageNet Sketch and ImageNet-R. Meanwhile, the average effective robustnesses of pre-trained models on in-support splits (orange) are consistently low.

examples have a class-specific tint while the remaining $20\%$ are randomly tinted. Meanwhile, in the shifted distribution, examples are always randomly tinted and are also padded.

To extrapolate to padded examples, we initialize a CLIP ViT-B/32 with zero-shot weights and fine-tune it on the reference distribution. To handle the spurious correlation between tint and label, we consider the intervention of training on randomly tinted examples, which we refer to as *balancing*.

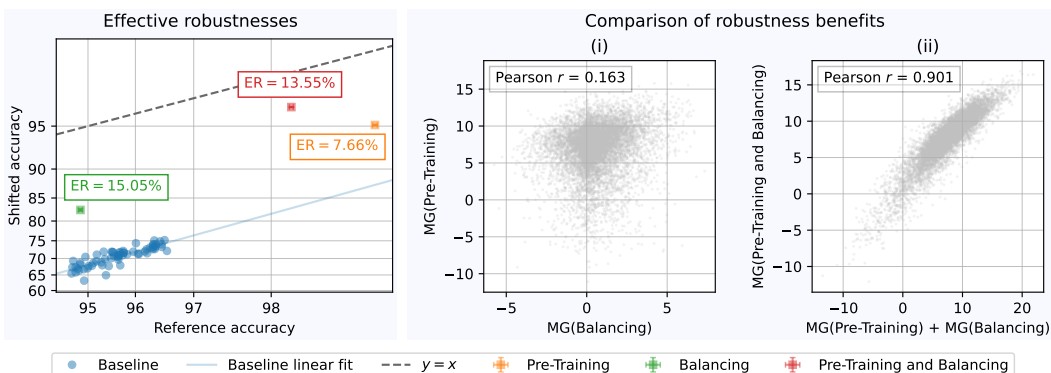

Figure 14: **Combining pre-training and balancing on a synthetic CIFAR-10 distribution shift.** Pre-training and balancing (an "oracle" intervention for handling dataset biases) each yield some effective robustness (ER) and combining these two interventions yields a high effective robustness and the highest shifted accuracy (left). The margin gains (MG) of pre-training and balancing on the shifted dataset are nearly uncorrelated (right, i), indicating that they improve performance on *different* subpopulations. Meanwhile, the margin gains of using both interventions are very highly correlated with the *summed* margin gains of using each individually (right, ii), suggesting that combining pre-training and balancing improves performance on *both* of these subpopulations. Error bars denote 95% confidence intervals over 64 random trials.

This is an "oracle" of sorts for handling dataset biases; it simply modifies the training distribution such that spurious features are not useful.

As with WILDS-FMoW, we find that pre-training and balancing each yield some effective robustness (see the left side of Figure 14). In this case, combining the two does not yield the greatest effective robustness, but does have the highest shifted accuracy. We apply the same methodology as in Section 5 to understand the robustness benefits of pre-training and balancing. We observe that their margin gains have a low positive correlation and that the margin gains of using both interventions are very highly correlated with the *summed* margin gains of using each individually (see the right side of Figure 14). These results corroborate our finding that pre-training and interventions designed to handle dataset biases can be complementary.

## C.4 CURATING DATASETS FOR FINE-TUNING

### C.4.1 UNDERSTANDING THE ROBUSTNESS BENEFITS OF PRE-TRAINING WHEN FINE-TUNING ON A CURATED DATASET

In Section 6, we find that fine-tuning on a curated dataset with only 64 examples can yield a performant and robust model for hair color classification. We observe that pre-training is necessary for effective use of the small curated dataset; in particular, training a model from scratch on a curated dataset yields robustness gains, but these gains are smaller and many more examples are required to attain comparable accuracy.

In this section, we shed additional light on how pre-training helps in this setting. Based on our intuition from Sections 3 and 4 that pre-training helps specifically with extrapolation, we hypothesize that pre-training provides two benefits when training on a small curated dataset. First, a pre-trained model may be able to extrapolate better from a small number of examples. This would result in both higher accuracy on the original CelebA distribution and higher worst-group accuracy, which we observe in Figure 6b. Second, recall that our curated dataset consists entirely of females, but hair color classification models are expected to perform well on males too. To compare different model's ability to extrapolate along this axis, we plot the balanced accuracy on males against the balanced accuracy on females. In Figure 6a, we observe that the pre-trained model indeed generalizes better to males than models trained from scratch.

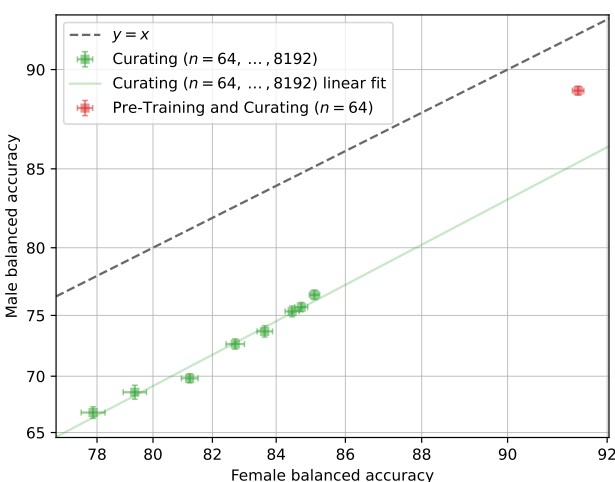

Figure 15: **Comparing extrapolation from females to males of pre-trained models and models trained from scratch.** We plot the balanced accuracy on males against the balanced accuracy of females of a pre-trained model fine-tuned on the curated dataset from Section 6 (red) and models trained from scratch on this dataset (green). Models trained from scratch establish a linear relationship between male and female balanced accuracy; however, the pre-trained model outperforms this trend, suggesting that it more effectively extrapolates to males from the female-only curated dataset.

# D  ADDITIONAL DISCUSSION

## D.1  ALTERNATIVE FINE-TUNING STRATEGIES

In this work, we focus on the common setting in which a pre-trained model is fully fine-tuned. It is important to note that pre-trained models used in a zero-shot context (i.e., without fine-tuning) and partially fine-tuned models (e.g., only the final classification layer is updated) are frequently more robust than fully fine-tuned models (Radford et al., 2021; Miller et al., 2021; Kumar et al., 2022). Such models may have higher effective robustness than fully fine-tuned models or in some cases may even outperform fully fine-tuned models on the shifted distribution. However, such models are typically less performant on the reference distribution than fully fine-tuned models.

Several works observe this tradeoff between performance on the reference distribution and robustness and devise methods for mitigating it, i.e., methods for *robust fine-tuning* (Wortsman et al., 2021; Hewitt et al., 2021; Kumar et al., 2022). For example, Kumar et al. (2022) argue that full fine-tuning "distorts" pre-trained features and propose linear probing *before* full fine-tuning (LP-FT) to prevent distortion. They also suggest that fine-tuning a model initialized as a zero-shot classifier may have a similar effect. In addition to full fine-tuning, in Section 4.1 we thus consider LP-FT and zero-shot initialization for fine-tuning. On in-support shifts, we observe that LP-FT and zero-shot initialization do not provide effective robustness benefits compared to full fine-tuning (see Figure 3), suggesting that these strategies do not help mitigate dataset biases.

Another strategy for robust fine-tuning is to ensemble a zero-shot model and a fully fine-tuned model. Both weight-space ensembles (Wortsman et al., 2021) and output-space ensembles (Hewitt et al., 2021) have been shown to improve robustness, sometimes even without sacrificing performance on the reference distribution. In fact, this strategy can yield robustness benefits even when dataset biases are a primary failure mode because the zero-shot model is independent of the biased reference dataset. Our work seeks to complement such empirically effective strategies by providing an understanding of when they are necessary. In particular, our findings suggest that ensembling is valuable precisely when dataset biases cause failures.

## D.2  CAN PRE-TRAINING HURT EXTRAPOLATION?

In this work, we discuss distribution shifts in which pre-training is beneficial to a model's ability to extrapolation outside of the reference distribution. A natural question to consider is whether pre-training can instead *hurt* it, yielding worse extrapolation than a model trained from scratch. A recent work by Salman et al. (2022) suggests that this is indeed possible. Specifically, they show that biases of pre-trained models can persist during fine-tuning. For example, a model pre-trained on ImageNet and fine-tuned on CIFAR-10 is highly sensitive to the presence of tennis balls (which are an ImageNet class but not a CIFAR-10 class). Meanwhile, a model trained from scratch on CIFAR-10 is not particularly sensitive to tennis balls. Thus, under a hypothetical "tennis ball shift" in which tennis balls appear in images in the shifted distribution, a pre-trained model would be less robust than a model trained from scratch. In this instance, pre-training provides a *harmful* prior for how to extrapolate.

## D.3  RELATING IN-SUPPORT AND OUT-OF-SUPPORT SHIFTS TO EXISTING CHARACTERIZATIONS

The characterizations relevant in this work, *in-support shift* and *out-of-support shift*, overlap with many existing definitions. Ye et al. (2022) introduce notions of *correlation shift* and *diversity shift* (closely aligned with in-support and out-of-support shifts, respectively) and provide a method for measuring the "amount" of each type of shift in a given distribution shift (similar to our method for dividing a distribution shift into in-support and out-of-support splits). Subpopulation shift (and its sub-types), shifts involving spurious correlations, covariate shift, and label shift are typically in-support. However, there are exceptions; for example, some works consider subpopulation shifts in which a subpopulation does not appear in the reference distribution (Santurkar et al., 2021; Yang et al., 2023), which are out-of-support. Domain generalization problems are nearly always out-of-support and extrapolating effectively outside of the reference distribution is often a key challenge of these tasks.

### D.4 UNDERSTANDING THE ROBUSTNESS OF PRE-TRAINED LANGUAGE MODELS TO SPURIOUS CORRELATIONS

Tu et al. (2020) study the robustness of pre-trained language models to distribution shifts with spurious correlations. Their central finding is that pre-training *can* improve performance on shifted datasets in which spurious correlations do not hold. They illustrate that this is because pre-trained models can generalize better from the small number of counterexamples to these correlations in the reference dataset. This is a similar phenomenon to our observation from Section 4.1 that pre-training can provide limited effective robustness even on in-support shifts (which we further explain in Section C.1.2). In cases such as those discussed by Tu et al. (2020), we hypothesize that pre-training can help to a limited extent by extrapolating better, but cannot mitigating the underlying failure mode of dataset biases.

### D.5 ADDITIONAL RELATED WORK

**Pre-training.** Pre-training a model (or taking an existing pre-trained model) and then fine-tuning it on a task-specific dataset is a common practice when developing machine learning models, often significantly improving performance over training a model from scratch (Sharif Razavian et al., 2014; Sun et al., 2017; Kornblith et al., 2019; Kolesnikov et al., 2019). Pre-training can be effective even when the downstream task is unrelated to the pre-training task, suggesting that pre-training yields useful general-purpose features; for example, object classification models trained on ImageNet (Deng et al., 2009) are good initializations for remote sensing (Xie et al., 2016) and medical imaging (Ke et al., 2021) tasks. Although greatly effective, pre-training is not without limitations. In some settings, pre-training does not improve performance over a randomly initialized model trained for long enough (He et al., 2019). Downstream performance can saturate as performance on the pre-training task improves (Abnar et al., 2021). Finally, biases of pre-trained models can persist after fine-tuning (Salman et al., 2022).

**Distribution shift robustness.** Machine learning models are often deployed in different environments from those in which they are trained. Such distribution shifts can cause models to significantly underperform (Koh et al., 2020; Gulrajani & Lopez-Paz, 2020; Hendrycks et al., 2020a). Numerous interventions have been proposed to improve the robustness of models, often targeting particular types of shifts. These include algorithmic interventions (Arjovsky et al., 2019; Byrd & Lipton, 2019; Sagawa et al., 2020a; Liu et al., 2021; Kirichenko et al., 2022; Idrissi et al., 2022) (often requiring group information), data augmentations (Hendrycks et al., 2020a; Goel et al., 2020) and pre-training (discussed below). However, interventions proposed thus far have failed to provide consistent benefits across distribution shift benchmarks (Koh et al., 2020; Gulrajani & Lopez-Paz, 2020; Hendrycks et al., 2020a; Wiles et al., 2021; Ye et al., 2022), rendering distribution shift robustness a persistent challenge.

