# OpenReview forum: "Ask Your Distribution Shift if Pre-Training is Right for You"
_ICLR.cc/2024/Conference — Submitted to ICLR 2024_

### Official Review · Reviewer_Uetj · 2023-10-31

**Soundness:** 3 good
**Presentation:** 2 fair
**Contribution:** 1 poor
**Rating:** 3
**Confidence:** 4

**Summary:**

The paper investigates why does initialization with a pretrained model improves performance on some tasks but not on others?
They argue that a model that is trained from scratch cannot perform well on examples that are not in the support of the training distribution, and argue pretraining can help on such out-of-support instances.
Pretraining, however, as they show cannot help with systematic biases (spurious correlations) in the training data. By combining initialization with a pretrained model and training with a balanced dataset that is free of spurious correlations, we get best of both the worlds.

The paper is mostly easy to follow but I was not surprised by their results, i.e. it did not improve my understanding of the problem or offer a novel practical advice.

**Strengths:**

- The focus of the paper is practically very relevant.
- The presentation is somewhat easy to follow

**Weaknesses:**

- **Unclear Contributions**. The contribution of the paper is unclear. Both theoretical and empirical contributions are mild. See next point for comment on theoretical analysis. The paper suggested to use initialization with a pretrained model and training on a balanced dataset, which is a standard practice anyway.
- **Limited theoretical contribution.** Theoretical analysis considered a very simple setting with an intuitive result. The result is too simple to inform poor extrapolation of random initialization (or training from scratch) or better extrapolation of pretraining in practice.
- **Vague or inconsistent definition of support**. Out-of-support and in-support are not formally defined. From their analysis and examples, the definition of out-of-support are examples with zero support (or probability) attributed by the input pdf. However, their experiments in Section 4.2 classified examples in and out-of-support using a classifier that is trained to classify between two distributions instead of estimating and using a PDF.
- **Presentation issues**. ER is a central measure used in the paper, but is not defined. How does the y axis label of Figure 5 (right) relate to MG defined in (4)?
- **Practical implications are not well argued**. Their takeaways may not be practically relevant. In practice, the distribution shift is likely a mix of "out-of-support" (whatever that means) and minority subpopulation. The takeaway of the paper that pretraining helps on out-of-support examples do not bear any practical significance in how we train and deploy a model on the target distribution.

**Questions:**

**Q1** In Figure 2 (left), if the training data only has pictures of _indoor dogs_ and _outdoor cats_, then are the examples of _outdoor dogs_ and _indoor cats_ inside or outside the support of the training distribution?

**Q2** In Figure 6(b), how does finetuning a pretrained model on female only dataset (i.e. female only examples from the original dataset) compare with the other results?

Minor: In figure 3, best to clarify that the cat and dog thumbnails are only placeholders and not from CIFAR-10.

---

> ### Author Response · Authors · 2023-11-21
> **Rebuttal Response**
>
> We thank the reviewer for the helpful feedback.
>
> **Unclear contributions**
>
> The reviewer states that “the paper suggested to use initialization with a pretrained model and training on a balanced dataset, which is a standard practice anyway.” While training on balanced data has indeed been suggested by previous work, our work (in Section 6) suggests that, as a principle, this balanced dataset can be small and even non-diverse when fine-tuning a pre-trained model. We believe that this intuition is valuable for informing dataset curation processes.
>
> Our work also offers an understanding of why pre-training in conjunction with a method for handling bias (such as balancing a dataset) is effective for robustness, which has not been previously studied to our knowledge.
>
> **Limited theoretical contribution**
>
> Please refer to the general response.
>
> **Vague or inconsistent definition of support**
>
> As noted by the reviewer, in-support and out-of-support examples are examples inside and outside the support of the reference distribution, respectively. In other words, an example is in-support if its probability density under the reference distribution $p_\text{ref}$ is small. To divide a shifted distribution into in-support and out-of-support splits, we instead estimate the probability density ratio $p_\text{ref}/p_\text{shift}$ and find examples where this quantity is small. The reviewer was confused that this is a different quantity than $p_\text{ref}$. As we explain and justify in Appendix B.3.1, we do so because estimating $p_\text{ref}$ is difficult for high-dimensional inputs (in our case, images) and $p_\text{ref}/p_\text{shift}$ is similarly meaningful.
>
> **Presentation issues**
>
> We are surprised by the reviewer’s comment that *effective robustness* (ER) is not defined, as it is defined in the  Section 2. To further clarify this metric, we added a longer definition in the Appendix in the revision.
>
> **Practical implications are not well argued**.
>
> Please refer to the general response.
>
> **Q1**
>
> If the training data only has pictures of indoor dogs and outdoor cats, then examples of outdoor dogs and indoor cats are out-of-support. Here, we expect that a model could suffer both from biases and from poor extrapolation to this previously unseen group.
>
> **Q2**
>
> We thank the reviewer for raising this point. In experiments that are not currently included in the paper, we found that fine-tuning a pre-trained model on a female only dataset is ineffective (the model still performs poorly on blond males). This might be because there are other attributes spuriously correlated with hair color even among just the female population, and this is why we opt for counterfactual editing. We will include this ablation in future revisions.
>
> We would like to note that we are not claiming that counterfactual data is the only or most effective way to de-bias a dataset. The purpose of this experiment is to illustrate that de-biasing a small dataset and fine-tuning a pre-trained model on it can be effective, regardless of the de-biasing method.

---

### Official Review · Reviewer_f9zp · 2023-10-31

**Soundness:** 1 poor
**Presentation:** 2 fair
**Contribution:** 2 fair
**Rating:** 3
**Confidence:** 4

**Summary:**

This paper investigates an interesting problem of the impact of pre-training data on the model's robustness to distributional shifts. The work proposes to characterize the "failure modes" of the model under distributional shifts into two types–poor extrapolation and dataset biases. The work then argues that pre-training can help with the first but not the second. This paper proposes two approaches to address these issues–use intervention techniques at pre-training to prevent exploiting biases; and fine-tune on small, non-diverse but debiased datasets.

**Strengths:**

The problem being investigated is definitely of interest. With the emergence of foundation models, it is of growing interest to better understand the impact of pre-training data and its implications for downstream processes.

This paper is well-contextualized. Its technical structure is plausible (intuitions, motivating examples, formal analysis, generalization, empirical verification, etc.).

**Weaknesses:**

This paper is a difficult read in general. I was quite attracted by the topic of this paper and had high hope until Theorem 1, which I could not understand after several attempts. **Many technical details are missing or inconsistent (e.g., missing key definitions, no details for important procedures, only providing references with no description at all), rendering many arguments ungrounded and hardly convincing.**

- Theorem 1, key error–$w_{ref}$ is undefined, which is a crucial variable. With this, I can only guess about this theorem. Assuming it is correct in its own sense, the conclusion is problematic. The theorem only gives that pre-training/initialization affects ID but not OOD. Yet, this affect can be either positive or negative,meaning pre-training/initialization does not necessarily help in every case–which I think is true in practical cases. This work has then taken it for granted that pre-training WILL HELP downstream robustness and built the arguments on this as a main basis. **I think there is a major gap.** It is necessary to specify the conditions when it helps/when it hurts/and when it does not affect, which is actually the most valuable part.

- Another fundamental issue for Theorem 1 is that the model is considered deterministc. Even if you use random initialization and stochastic gradient solvers, you end up with the same solution. It is natural to use a stylized model as a starting point to build the analysis. Then I would expect to see how this analysis generalizes to the case of non-convex models where there is inherent learning stochasticity. Yet, without providing anything else, the paper jumps to experiments that are all based on neural network models. **This is another major gap.** Actually, there is not really a "pre-training" thing for convex models–regardless of the order you feed data to the model, it always converges to the same optimal solution. The resulting model solely depends on the training data and is irrelevant to initialization or the training process.

- I don't understand why the proposed "in-support shifts" would change the classification boundary at all. Image a max-margin classifier (e.g., SVM) and a binary classification task for cat and dog images. (Note that Logistic Regression requires the same probability for both classes. You cannot directly apply it to unbalanced classification problems.) Regardless of the relative proportion for cat or dog images, the underlying distribution for cats and dogs is invariant. A proper classifier would find the decision boundary somewhere between the generating distributions for both classes. I don't see why this is considered "not robust". Or a more important question–what is the robustness considered in this work? **The definition for the central notion of this work is not provided.**

- I don't understand the splitting methods for partitioning sketch images into ID and OOD subsets w.r.t. ImageNet. The paper only describes it based on whether they "look like". This process actually sounds rather non-trivial. The paper refers to Appendix B 3.2 for details, **which does not exist.**

- For the proposed approaches–use intervention techniques at pre-training to prevent exploiting biases; and fine-tune on small, non-diverse but debiased datasets. The paper merely cites existing works for these techniques without any description. **This renders this work not self-contained and incomplete and this cannot be counted as technical contribution of this work.**

- Format: Appendix is not cut from the main paper. The PDF provided for the main paper is this 32-page document.

**Questions:**

- Theorem 1, key variable $w_{ref}$ is undefined.

- Appendix B 3.2, which is referred to in Page 6, does not exist.

- Appendix should not be submitted under the main paper.

---

> ### Author Response · Authors · 2023-11-21
> **Rebuttal Response**
>
> We thank the reviewer for the helpful feedback.
>
> **Concerns about Theorem 1**
>
> The reviewer expressed confusion about $w^*_\text{ref}$ being undefined. We state that $w^*_\text{ref}$ does not depend on the initialization $w_\text{init}$. We also state that it is a property of the reference dataset. In the revision, we clarify this when we first refer to $w^*_\text{ref}$.
>
> The reviewer correctly points out that pre-training can in fact hurt robustness, which we discuss in Appendix D.2. Indeed, this is reflected by our theoretical analysis, which suggests that pre-training can influence a model’s extrapolation, not that it will necessarily help (this is why we use the language “can help with extrapolation”). See the general response for more discussion.
>
> While the model in Theorem 1 is deterministic, crucially, it depends on initialization (in our case, whether the model is pre-trained or randomly initialized). Even though the loss is convex, there are multiple equally good solutions and the initialization specifies which is selected. Pre-training thus can affect the learned model, even in this convex setting.
>
> See the general response for concerns about the simplicity of the theoretical analysis.
>
> **Robustness to in-support shifts**
>
> We are confused by the reviewer’s comments about models being agnostic to in-support shifts. Consider [1], for example.
>
> **Definition of robustness**
>
> In our work, we give both an intuitive and formal definition of robustness. Intuitively, as described in our introduction, we consider the robustness of a method to be its ability to succeed on a shifted distribution that differs from the reference distribution the model is trained on.
>
> Formally, we define and evaluate the robustness of a model via the *effective robustness* metric [2] (we define this metric in Section 2). To further clarify this metric, we added a longer definition in the Appendix in the revision.
>
> **Splitting Method**
>
> We are unsure why the reviewer is unable to view Appendix B.3.2. We have opened and download the pdf on multiple browsers, and in each case the Appendix is included in the file. We also note that, as we state in the main text in Section 4.2, we actually give the full details in the splitting method in Appendix B.3.1 and not B.3.2.
>
> **Using Interventions to prevent exploiting biases**
>
> We would like to point the reviewer to Appendix B.4. Here, we indeed give a description of the *Deep Feature Reweighting* (which we assume the author is referring to) and also carefully describe our implementation of this method. Our contribution here is not a specific approach, but rather a guiding principle of combining pre-training with an intervention that can handle biases. Thus, we did not consider the details of the specific choice of this intervention to be worth including in the main paper.
>
> **Location of the Appendix**
>
> We refer the reviewer to the ICLR 2024 call for papers and author guidelines:
>
> - The ICLR 2024 author guide states: “We encourage authors to submit a single file (paper + supplementary text). Please mark the supplementary material clearly.”
> - The ICLR 2024 call for papers states: “Authors may use as many pages of appendices (after the bibliography) as they wish.”
>
> [1] Chaudhuri et al. “Why does Throwing Away Data Improve Worst-Group Error?” (2023).
>
> [2] Taori et al. “Measuring Robustness to Natural Distribution Shifts in Image Classification.” (2020).

---

> > ### Comment · Reviewer_f9zp · 2023-11-23
> >
> > Thanks to the authors for the rebuttal. I have read it in full. My main concern about the theorem and conceptual results remains. I cannot agree with the arguments on convex model having multiple equally good solutions–this only happens in degenerate cases (consider linear program as an example). It is unlikely for convex optimization problems with continuous variables (excluding linear programs) to have multiple GLOBAL optimal solutions which have exact same optimal values. More often than not, multiple optimal solution only happens when the final solution is on the decision boundary–in which case the results are often trivial.
> >
> > The case with multiple optimums is for nonconvex optimization, where one cannot find a global optimum in polynomial time and has to reduce to LOCAL optimums. Which local optimum one ends up with will depend on its initialization and solution trajectory.

---

### Official Review · Reviewer_cFpN · 2023-11-01

**Soundness:** 2 fair
**Presentation:** 2 fair
**Contribution:** 2 fair
**Rating:** 3
**Confidence:** 3

**Summary:**

The paper proposes to study the effectiveness of model pre-training under various kinds of distribution shifts. A key insight of the paper is that pre-training can help address poor extrapolation but not dataset biases. The paper motivates this insight theoretically and demonstrates the expected behavior empirically in a variety of experimental setups. These include simulated synthetic and real-life shifts, as well as a case study on dataset de-biasing.

**Strengths:**

- The paper addresses the very interesting topic of exploring whether pre-training helps training performant models under different types of distribution shift. This is an important frontier in the increasingly adopted pre-training fine-tuning training methodology.
- The ideas presented seem original and the relevant literature is sufficiently discussed.
- The later sections on developing more robust models were interesting case studies on how to apply the insights from previous sections.

**Weaknesses:**

- Although more details are presented in the Appendix, I don't think that the setup for Theorem 3.1 is presented well in the paper. It is not clear why the logistic regression assumption is needed and the proof is not referenced in the main paper. It is also unclear what $proj_{W_{ref}}$ refers to as it is never formally introduced. Is orthogonal to be interpreted mathematically or figuratively? Also "[...] while the initialization determines how the model extends outside of $W_{ref}$": since $W_{ref}$ is contained in $proj_{W_{ref}}$. It appears to me like both terms are influenced by $W_{ref}$, not just the first term.
- Figure 1's legend is unreadable due to overlapping text. I was still able to get the intuition but the authors should fix this.
- Although Figure 3 presents two examples of in-support and out-of-support shifts, there is no ablation on the shift intensity. I wonder to what extent the shift type influences shifted accuracy and how the shift intensity for a fixed type of shift would alter the experimental results. In other words: does the strength of the bias or the degree of extrapolation matter? My intuition is that this should also play a role. Connecting to this, both in-support and out-of-support shifts are also not formally defined. The descriptions given at the beginning of section 4 should be made more precise.
- The negative correlation reported in the middle panel of Figure 5 is negative but at the same time the correlation of -0.112 is weak, suggesting that the margin gains are independent of each other rather than complimentary.
- Overall, the take-away message from this work is a bit unclear to me. While the main paper suggests that pre-training is always desirable (with larger gains for out-of-support shifts than for in-support shifts), Appendix D.2 discusses the possibility of harmful biases being instilled into the model during pre-training. Without assumptions about which distribution to expect at test time (which is what we would want to fine-tune for), it therefore becomes impossible to understand whether a model should have been pre-trained or not. This makes it hard for this method to be applied in practice.

**Questions:**

Embedded in Weaknesses above.

I am willing to increase my score as part of the discussion phase if the authors can address my concerns.

---

> ### Author Response · Authors · 2023-11-21
> **Rebuttal Response**
>
> We thank the reviewer for the helpful feedback. Please see general response for a discussion on:
>
> - Takeaways
> - Cases where pre-training hurts robustness
> - Correlation of margin gains
>
> Below, we address the reviewer’s remaining concerns:
>
> **Notation in Section 3**
>
> To clarify the reviewer’s confusion about Theorem 1:
>
> 1. $proj_{W_\text{ref}}$ is the projection operator onto the subspace $W_\text{ref}$. We apologize for the confusion and clarify this in the revision.
> 2. Orthogonal is to be interpreted mathematically; $w_\text{init}-proj_{W_\text{ref}}w_\text{init}$ is orthogonal to $W_\text{ref}$ because we remove the component that lies within $W_\text{ref}$. This is also why this second term would not influence the model’s output on any point within $W_\text{ref}$ (because the product of this point and $w_\text{init}-proj_{W_\text{ref}}w_\text{init}$ would be zero).
> 3. The proof is in Appendix A and is referenced in the paper.
>
> **Overlapping text in Figure 1**
>
> We are confused by the author’s comments about the overlapping text in Figure 1. We were unable to reproduce this issue when
>
> 1. Downloading the PDF and viewing it locally.
> 2. Viewing the PDF in Safari, Chrome, Brave.
>
> We suggest that the reviewer uses one of the methods above to view the PDF.
>
> **Ablations on shift intensity**
>
> We thank the reviewer for the suggestion of ablating shift intensity, and intend to include this in the Appendix in future revisions. We already have some intuition of how ablating the fraction of samples from the minority group (e.g., randomly tinted examples in the tint shift) affects the effective robustness of pre-trained models. In our existing experiments, we observe that when the number of samples available from minority groups is small, pre-trained models do have a little effective robustness, even under in-support shifts. However, in Appendix C.1.2, we provide evidence that as we increase the total number of samples (while maintaining the same distribution), these effective robustness benefits vanish. If we were to decrease the fraction of samples from the minority groups while maintaining the total dataset size, we would expect the effective robustness of pre-trained models to increase somewhat, but as we discuss in Appendix C.1.2, this robustness could be attributed to better extrapolation from a small number of samples.

---

> > ### Comment · Reviewer_cFpN · 2023-11-23
> > **Thank you**
> >
> > I thank the authors for their rebuttal which has clarified some of my concerns. However, I am unfortunately still not convinced that the presented characterization of in-support and out-of-support shifts are the main directions on which to categorize whether pre-training is helpful or not. As a result, I will not be able to change my score.

---

### Official Review · Reviewer_kid6 · 2023-11-01

**Soundness:** 1 poor
**Presentation:** 2 fair
**Contribution:** 2 fair
**Rating:** 3
**Confidence:** 4

**Summary:**

The paper studies the robustness of classifiers, based on pre-training, to distribution shifts. It provides with a chacterization for distribution shifts based on whether the network is asked to extrapolate (out-of-support) or generalize in a group-balanced manner when trained with data which are label-imbalanced or are generated from spuriously correlated features (in-support). They claim that pretraining helps with the former type, but not the latter; providing some theoretical insights in a very simplified setting and performing various ablating experiments. In order to deal with the second type, they argue that pretraining needs to be combined with group-robustness methods, such as methods based on loss reweighting or data rebalancing; and that this strategy provides with complementary robustness benefits (to both proposed types). They conclude by encouraging “practitioners not to treat pre-training as a panacea for robustness”.

**Strengths:**

The paper is easy to follow and for the most part well-written. While the paper does not propose novel methods for robustness, it leads an important discussion on the interplay of pre-trained networks with training methods for group-robustness. It provides with a novel characterization of distribution shifts and it argues about the complementary utility of the two approaches around this characterization, with some theoretical and empirical insights. The reviewer believes that such an analysis paper is needed in light of recent advances in the studied literature and that the authors have identified well that this is a topic of interest.

The reviewer appreciates the careful discussion in defining and designing in-support and out-of-support shifts, and they believe it is a potentially useful axis of discussion.

**Weaknesses:**

While their conclusion might not be incorrect, the design of several experiments is flawed and does not lead to the authors’ claims. The reviewer thinks that these consist a large enough body of the paper to lean towards possible acceptance. In particular:

1.  In **Section 4/Figure 3**, there are multiple variables which are being ablated at the same time. The measured effective robustness is with respect the performance of ResNet18 models, however the pretrained models, which are compared to, have various architectures. It is not clear whether the perceived robustness is due to purely transfer properties from pretraining strategies or from larger model size.
2. Biased datasets are also susceptible to an *in-support*/*out-of-support* analysis, however one is not given at the study. Specifically, one can imagine the extreme case where some minority group probabilities go to 0, yielding them completely *out-of-support*. For example, imagine the case in **Figure 2 (right)**, where cats are also observed during the night. How does this impact the pretraining algorithms? Current analysis seems to suggest that pretraining would be able to handle the more extreme cases better, which is counter-intuitive. I suggest studying systematic generalization tasks, where some combinations of generative attributes are completely not represented in the training set, but they exist in the test set in a balanced manner. See [1,2].
3. In **Section 5/Figure 5**, the first correlation estimate is not strong enough to indicate negative correlation of (i) (and thus complementary effects between pretraining and debiasing methods). Please provide with a confidence interval of the correlation estimate, as it seems statistically possible that it is very close to 0. On the other hand, the correlation of (ii) is poorly motivated and it can be tautologically positive, in which case it is of no logical inference value.
4. In **Section 6**, the proposed form of curated dataset is made so that the spurious feature which is balanced during test-time is completely omitted, namely the “gender” attribute of CelebA. If there is no spurious correlated feature during training, then the transfer problem corresponds to an extrapolation one, for which probably a classifier with simple augmentation/regularization might just work. The construction of counterfactual data, beyond being difficult to achieve in most cases, needs to be ablated to demonstrate that it is a necessary component of the curation process.

Finally, a comparatively minor concern is that the theorerical insight provided corresponds to an overly simplified setting.

[1] Schott, Lukas, et al. "Visual representation learning does not generalize strongly within the same domain." (2021).
[2] Tsirigotis, Christos, et al. “Group Robust Classification Without Any Group Information.” (2023).

**Questions:**

In **Figure 4**, what % of samples are found to be *in-support* for each of the test sets considered? How is absolute test accuracy (iid and ood) affected as we ablate this? In other words, would a test set comprised 100% of *in-support* samples achieve similar absolute accuracy as an iid test set?

---

> ### Author Response · Authors · 2023-11-21
> **Rebuttal Response**
>
> We thank the reviewer for the helpful feedback.
>
> **Measuring the robustness of pre-trained models with various architectures with respect to ResNet-18 models trained from scratch (1)**
>
> The reviewer expressed concerns that we measure the robustness of pre-trained models with various architectures with respect to ResNet-18 models trained from scratch. We would like to point out that the pre-trained models include ResNet-18s (pre-trained on ImageNet) in addition to other, larger architectures, so there is already a direct comparison between models with the same architecture. That said, in future revisions we will augment this result by adding models trained from scratch with ResNet-50 and DenseNet-121 architectures (we also tried ViTs, but these tend to perform poorly when trained from scratch on small datasets [1]).
>
> **Biased datasets with zero minority sub-group probabilities (2)**
>
> We thank the reviewer for pointing out that there might also be biased datasets in which the minority group probabilities are close to zero. This is an important point that we intend to clarify in future revisions. Indeed, in this extreme case a model would suffer from the bias. However, a model could also suffer from poor extrapolation, since the minority groups represent a previously unseen setting. We do not categorize such a shift as in-support for this reason.
>
> The reviewer says that “current analysis seems to suggest that pre-training would be able to handle the more extreme cases better, which is counter-intuitive”. We agree that this result as stated by the reviewer would be counter-intuitive; however, this is not what our current analysis suggests. Our analysis suggests that pre-training can help with extrapolation, implying that a pre-trained model might be more robust *than a model trained from scratch under the same extreme case*. Notably, the analysis does not suggest that a pre-trained model would perform better in this extreme case than in the case where data from the minority group is available (since, indeed, a pre-trained model would likely suffer more from biases).
>
> **Margin gain correlations (3)**
>
> In the general response, we discuss why, in our view, even uncorrelated margin gains imply that interventions are complementary.
>
> We are confused about why the reviewer believes that the correlation of (ii) is poorly motivated and might be tautologically positive. In our understanding, for something to be tautologically true means that they mean the same thing. In this case, applying two interventions separately and summing their independent gains and applying them together and measuring the gain are not the same. While some correlation is to be expected, the high positive correlation provides evidence that applying the two interventions together yields the combined benefits of applying each individually.
>
> **Counterfactual data in the curation process (4)**
>
> The reviewer points out that “if there is no spurious correlated feature during training, then the transfer problem corresponds to an extrapolation one”. This is exactly the purpose of the dataset curation process; since our analysis suggests that pre-training can help with extrapolation, if we can set up a dataset where the problem is just extrapolation then a pre-trained and fine-tuned model might do well. While there may be other strategies besides pre-training that can help with extrapolation, we do not believe that “simple augmentations/regularization” would be an effective solution in general.
>
> We are not claiming that counterfactual data is the only or most effective way to de-bias a dataset. The purpose of this experiment is to illustrate that de-biasing a small dataset and fine-tuning a pre-trained model on it can be effective, regardless of the de-biasing method. We intend to clarify this in future revisions.
>
> **Question about the in-support and out-of-support splits in Figure 4**
>
> We report the number of samples found to be in-support and out-of-support in Table 1 in Appendix C.2.1. As the reviewer alludes to, the accuracy on the in-support split is higher than the average accuracy under distribution shift (since the in-support split contains “easier” examples). We conduct an analysis in Appendix C.2.4 illustrating that, even accounting for this difference in difficulty between the in-support and out-of-support splits, pre-training can help substantially out-of-support but not in-support.
>
> [1] Dosovitskiy et al. “An Image is Worth 16x16 Words: Transformers for Image Recognition at Scale.” (2021).

---

> > ### Comment · Reviewer_kid6 · 2023-11-22
> > **Answer to Rebuttal**
> >
> > **Measuring the robustness of pre-trained models with various architectures with respect to ResNet-18 models trained from scratch (1)**
> >
> > Thank you for acknowledging that. I am looking forward to further experiments using the same models (and generally ablating one variable at the time)!
> >
> > **Biased datasets with zero minority sub-group probabilities (2)**
> >
> > One can imagine “interpolating” between the case of systematic generalization (suppose $\lambda = 0$ - where training prob. of some minority groups is exactly zero) and fully unbiased in their factors training sets (suppose $\lambda = 1$ - where all groups are equally probable in the training set). Everything else in between ($\lambda \in (0, 1)$) corresponds to a training set generated by spuriously correlated attributes. We can see that for $\lambda > 0$ generalization is in-support, and the extreme case $\lambda = 0$ generalization is out-of-support. I would like to see the effective robustness that pretraining (ideally with the same training distribution and model) can provide as $\lambda \to 0$. I mentioned that according to the insights of your study, effective robustness should increase as we get closer to out-of-support (systematic) generalization; which to me seems as counter-intuitive.
> >
> > **Margin gain correlations (3)**
> >
> > Authors are agreeing by saying “While some correlation is to be expected“, to which I ask “how much and why?”.
> >
> > **Counterfactual data in the curation process (4)**
> >
> > There seem to be two independent processes here. One is the agressive filtering, the other is the counterfactual. They need to be ablated separately, in order to assess their utility even as practically unattainable, but theoretically interesting, constructs.
> >
> > **Question about the in-support and out-of-support splits in Figure 4**
> >
> > Thank you for your answer!
> >
> > Overall, while I appreciate the time you have taken to respond to the review, I choose to maintain my original assessment.

---

> > > ### Author Response · Authors · 2023-11-22
> > > **Rebuttal Response**
> > >
> > > We thank the reviewer for their response! We would like to reiterate points which the reviewer may have overlooked in our previous response.
> > >
> > > **Biased datasets with zero minority sub-group probabilities (2)**
> > >
> > > We thank the reviewer for elaborating. We would like to point out again that effective robustness is defined as robustness **beyond the baseline of models trained from scratch on the same reference data**. So, it is actually not counter-intuitive that "effective robustness should increase as we get closer to out-of-support (systematic) generalization". In the case of $\lambda=.2$ (which is the setup of our tint shift experiment, pre-trained models have very little effective robustness). In the out-of-support case of $\lambda=0$, one might imagine that both models trained from scratch and pre-trained models perform worse under distribution shift in absolute terms compared to $\lambda=.2$, but that pre-trained models do exhibit effective robustness above models trained from scratch. This effective robustness might stem from them extrapolating better to the previously unseen minority group than models trained from scratch, even though they still suffer from biases.
> > >
> > > **Counterfactual data in the curation process (4)**
> > >
> > > We would like to reiterate that the purpose of this experiment is to illustrate that de-biasing a small, non-diverse dataset and fine-tuning a pre-trained model on it can be effective, **regardless of the de-biasing method**. Hence, the details of the de-biasing method (in this case, the choice to sample 32 examples and create counterfactuals for these), is not directly relevant to this point.

---

### Author Response · Authors · 2023-11-21
**Rebuttal Response**

We thank the reviewers for their valuable feedback and questions. We respond to the reviewer’s concerns individually, and include responses to common concerns here.

**Takeaways and practical implications**

The primary contribution of our work is an understanding of the types of failures that (under full fine-tuning) pre-training can and cannot address. Reviewers point out that practitioners might not know what sort of distribution shifts they would encounter, and are confused by how this understanding might be operationalized. We offer initial explorations in Sections 5 and 6. For example, the approach of combing pre-training with interventions for mitigating bias is guided by this understanding, and results in models that address multiple types of failures. In particular, such an approach would be effective even without knowing the nature of the distribution shift in advance (as suggested by reviewer cFpN) or if multiple failure modes occur simultaneously (as suggested by reviewer Uetj).

**Pre-training hurting robustness**

A few of the reviewers pointed out that in some cases, pre-training might actually hurt robustness. Indeed, we discuss exactly this issue in Appendix D.2. We would like to note that we do *not* study when a particular pre-trained model *will* improve robustness but rather when pre-training *can* and *cannot* improve robustness. Stated otherwise, we are answering the following question:

“As pre-trained models grow larger and are trained on more/better data, are there limits to the type robustness that pre-training will be able to provide?”

rather than

“Will a particular pre-trained model help me with a particular distribution shift?”

We thank the reviewers for pointing this out and intend to clarify this in future revisions.

**Simplicity of the theoretical analysis**

We acknowledge that, as pointed out by reviewers, the theoretical analysis is conducted in a very simple setting. We provide this simple theoretical analysis to build intuition for why pre-training can improve robustness to some distribution shifts, but not others. We do not claim that this explains the robustness properties of large models and refer to our empirical results in these more practically relevant settings.

**Weak negative correlation of margin gains**

We thank reviewers for pointing out that a weak negative correlation of margin gains indicates uncorrelated benefits rather than complementary benefits. In our view, given that different models often make highly correlated mistakes [1], even uncorrelated margin gains would imply that interventions are, in a sense, complementary (since the baseline expectation would be that their benefits are highly correlated). That said, we understand the reviewers’ confusion and intend to clarify this in future revisions.

[1] Mania et al. “Model similarity mitigates test set overuse.” (2019).

---

### Meta-Review · Area_Chair_247r · 2023-12-02

**Metareview:**

Reviewers generally found the motivation of the paper interesting, and consider this direction to be worth further study. However, there were a number of concerns around clarity of the presentation (particularly around the main theoretical results), the design of the experiments, the simplicity of the theoretical setup, and the overall takeaways from the work. The authors provided a detailed response which helped clarify some points (e.g., takeaways); however, all reviewers maintained concerns (e.g., whether the in- versus out-support dichotomy fully captures the value of pre-training). We believe the paper would be best suited by incorporating changes following the reviewers' suggestions, and undergoing a fresh round of review.

**Justification For Why Not Higher Score:**

Unanimous recommendation that the paper needs more work, with concerns on clarity, experimental design, theoretical setup, and take-home messages.

**Justification For Why Not Lower Score:**

N/A

---

### Decision · Program_Chairs · 2024-01-16

Reject